# Polymers in High-Efficiency Solar Cells: The Latest Reports

**DOI:** 10.3390/polym14101946

**Published:** 2022-05-11

**Authors:** Paweł Gnida, Muhammad Faisal Amin, Agnieszka Katarzyna Pająk, Bożena Jarząbek

**Affiliations:** 1Centre of Polymer and Carbon Materials, Polish Academy of Sciences, 34 M. Curie-Sklodowska Str., 41-819 Zabrze, Poland; mfaisal-amin@cmpw-pan.edu.pl; 2Institute of Chemistry, University of Silesia, Szkolna 9, 40-006 Katowice, Poland; agpajak@us.edu.pl

**Keywords:** photovoltaics, dye-sensitized solar cells, bulk-heterojunction solar cells, perovskite solar cells, polymers, thin layers

## Abstract

Third-generation solar cells, including dye-sensitized solar cells, bulk-heterojunction solar cells, and perovskite solar cells, are being intensively researched to obtain high efficiencies in converting solar energy into electricity. However, it is also important to note their stability over time and the devices’ thermal or operating temperature range. Today’s widely used polymeric materials are also used at various stages of the preparation of the complete device—it is worth mentioning that in dye-sensitized solar cells, suitable polymers can be used as flexible substrates counter-electrodes, gel electrolytes, and even dyes. In the case of bulk-heterojunction solar cells, they are used primarily as donor materials; however, there are reports in the literature of their use as acceptors. In perovskite devices, they are used as additives to improve the morphology of the perovskite, mainly as hole transport materials and also as additives to electron transport layers. Polymers, thanks to their numerous advantages, such as the possibility of practically any modification of their chemical structure and thus their physical and chemical properties, are increasingly used in devices that convert solar radiation into electrical energy, which is presented in this paper.

## 1. Introduction

Given the ever-increasing demand for electricity and environmental pollution, new, efficient, and, very importantly, environmentally friendly sources of renewable energy are being sought. However, it is worth remembering that a very large amount of electricity is still obtained from fossil fuels; thus, to find alternatives to non-renewable fuels, very efficient and relatively cheap sources of green energy are needed. One of the fastest-growing branches of renewable energy sources is solar energy, specifically photovoltaics. It is worth remembering that solar energy can be used in two ways, not only by photovoltaic cells but also by solar collectors for, among other things, heating [1]. Solar cells are divided into three generations. The first generation is made up of crystalline silicon cells, which are currently the most commercially used [2]. Thin-film devices based on the CdTe, CIGS (Copper Indium Gallium Selenide), GaAs, and a-Si, among others, represent the second generation [3]. Currently, the third generation of solar cells, which includes dye-sensitized solar cells (DSSC), perovskite solar cells (PSC), and bulk-heterojunction solar cells (BHJ), among others, is the most widely researched and rapidly developed [4,5,6]. Of course, as it is well known, currently, the highest solar-to-electricity conversion efficiencies are demonstrated by multi-junction solar cells overcoming the 47% threshold. However, these devices are very expensive to prepare, and their processes are extremely difficult and complicated, which significantly limits the possibility of their commercial application [7]. The use of organic compounds in photovoltaic devices offers great opportunities through a wide range of possibilities to modify the chemical structure of these compounds and, consequently, change their physical and chemical properties. In addition, it is also worth mentioning their significantly lower production costs, less energy consumption, and simpler preparation methods [8,9,10,11]. Research in recent years has confirmed a significant increase in the energy conversion efficiency of third-generation cells. According to the literature reports, perovskite solar cells achieve efficiencies of over 25% [12], DSSCs over 14% [13,14] and BHJs around 18% [15]. Furthermore, a great advantage of organic compounds is that they can be applied to various substrates using various methods. Numerous attempts have been made with often positive results to prepare flexible photovoltaic cells with polymer substrates. Both BHJ [16,17,18], PSC [19,20,21] and DSSC [22,23,24] structured devices are widely used for the preparation of flexible solar cells when new methods of preparing and applying materials to polymer substrates are sought.

In recent years, huge interest in using new polymeric materials in organic photovoltaics (OPV) has emerged. In each of these three types of the third generation of solar cells, polymeric materials find a variety of very important applications. Considering each of the components of solar cells, one can multiply the associated application of polymer materials.

The layers of polymeric materials play different roles starting with dye-sensitized solar cells, which are characterised by their layered structure. The first layer of a DSSC is the substrate. The glass or polymer substrate is covered with a transparent conductive oxide (TCO); very often, fluorine-doped tin oxide (FTO) or indium tin oxide (ITO) are used. However, research on flexible photovoltaic cells is increasingly reported in the literature. The most common flexible substrates used in photovoltaics are made of polymers such as polyethylene naphthalate (PEN) or polyethylene terephthalate (PET) [22,23,25,26,27,28,29]. Subsequently, polymers are used as materials responsible for forming the porous structure of a semiconducting oxide layer, e.g., TiO_2_. For this purpose, polymers or copolymers such as polystyrene (PS), polyvinylpyrrolidone (PVP), P123 Pluronic (PEO_20_–PPO_70_–PEO_20_) is a triblock copolymer, copolymer of poly(vinyl chloride) (PVC) and poly(oxyethylene methacrylate) (POEM) [30,31,32,33]. Polymers are also widely used in the electrolyte layer. Currently, DSSCs containing a liquid electrolyte containing redox pair are very widely used; however, due to quite significant limitations resulting from the use of a liquid electrolyte, quasi-solid state DSSCs (qs-DSSCs) and solid-state DSSCs (ss-DSSC) are being developed. The polymers used include polyacrylonitrile (PAN), polyethylene oxide (PEO) or poly(ethylene glycol) (PEG) [34,35,36]. The last layer forming the DSSC is the counter-electrode, which is now often made of polymeric materials or composites thereof. These materials significantly reduce the cost compared to a platinum electrode. Common polymers used are polyaniline (PANI), polypyrrole (PPy), polythiophene (PTh) and its derivatives [37,38,39,40,41]. 

Polymeric compounds are also widely used in the BHJ solar cells, where the active layer is a mixture of donor and acceptor (D-A) materials. π-conjugated polymers are the most commonly used as electron-donor materials in the active layer, among others [42,43,44,45,46]. However, they are present less frequently as an acceptor among others [47,48,49]. Furthermore, polymers are also used as buffer layers to form a barrier between the active layer and the electrode. This barrier impedes the fast charge transfer, which leads to serious charge accumulation at the contacts. The charge accumulation increases the probability of recombination and deteriorates the performance of the device. It is, therefore, necessary to speed up the charge separation process and increase the transfer efficiency. 

Polymers are also being investigated for use in perovskite solar cells due to their diverse characteristics. To improve the nucleation and crystallization processes in the perovskite layer(s), polymers are added as additives [50,51,52,53]. Because of their proper charge mobility and energy level organization, polymers can also be utilized as electron and hole-transporting materials [54,55,56,57,58,59,60], as well as an interface layer, to prevent recombination and improve carrier separation efficiency [61,62,63,64].

Herein, we present the latest reports on polymeric materials used in photovoltaic solar cells. In our paper, three types of solar cells: dye-sensitized, bulk heterojunction and perovskite solar cells, are presented in three successive chapters, where the role of polymers and polymers thin films are described and discussed. Based on the latest literature reports, the photovoltaic parameters, such as open circuit voltage (V_oc_), short-circuit current (J_sc_), fill factor (FF) and power conversion efficiency (PCE) are gathered and compared for these types of solar cells.

## 2. Polymers in Dye-Sensitized Solar Cells (DSSCs)

DSSCs, increasingly studied worldwide, show an ever-increasing radiation conversion efficiency. A distinction is made between solar cells containing a liquid or gel electrolyte of a similar design and a solid electrolyte. Research on DSSCs began with the use of a liquid electrolyte, and it is this type of cell that is most commonly reported in the literature. This is mainly due to the fact that the preparation of a device with this structure is the least complicated and time-consuming and is ideal when testing new dyes. Additionally, the use of liquid electrolyte, although it improves the photovoltaic parameters of the device, has many disadvantages, among which we can specify the limitation of the temperature range of the device, problems with proper sealing of the solar cell and causing corrosion of materials. Hence, the idea of replacing the liquid electrolyte with a solid or gel electrolyte emerged. When using these two types of electrolytes, it is difficult to achieve the same high parameters as with a liquid electrolyte. Still, numerous reports in the literature emerge to suggest this is being achieved. Replacing the liquid electrolyte, especially with a gel electrolyte, significantly increases the stability of the device over time and the charge mobility is definitely higher than with solid electrolytes. As also mentioned in the introduction, polymers are increasingly used in solar cells with DSSC structures. The following subsections describe the roles they play in them, including polymeric dyes, which are very difficult to find in other review papers. The structures of these devices are shown in Figure 1a, while the chemical structures of polymers used in this type of solar cell are presented in Figure 1b.

### 2.1. Polymers as a Flexible Substrates

The commercial application of DSSCs is currently being intensively developed, and they are often used in glass façades that can additionally generate electricity. However, flexible substrates are increasingly being used to be able to make the whole device more flexible and thus greatly expand the application possibilities of this type of solar cell. In laboratory work, fluorine-doped tin oxide coated glass is used, which is being replaced by polymeric materials such as polyethylene terephthalate or polyethylene naphthalate deposited on ITO [22,65,66]. The previously mentioned ITO-embedded polymers are the most commonly used due to their numerous advantages, such as high transparency, low preparation costs, ability to form the required shapes, low weight, flexibility and low resistance [67]. When using flexible substrates, it is very important to prepare a conductive oxide layer such as TiO_2_ in a low-temperature manner in the region of 120–150 °C [66]. The resulting parameters for flexible DSSC, such as open-circuit voltage, short-circuit current, fill factor and power conversion efficiency over the last few years, are summarised below, in Table 1. The commercial dye di-tetrabutylammoniumcis-bis(isothiocyanato)bis(2,2′-bipyridyl-4,4′-dicarboxylato) ruthenium(II), denoted as N719, was used to prepare the devices. 

In [26], the preparation method for flexible photoanodes using the polyethylene naphthalate deposited on ITO was reported. The preparation of titanium dioxide-graphene quantum dot (TG) was used as a light-scattering layer (LSL) using facile electrodeposition and drop-casting. The photoanode denoted as ITO/PEN + T LSL was prepared by the facile electrodeposition from an aqueous solution of 0.1 M titanium tetraisopropoxide in 0.1 M LiClO_4_. The photoanodes containing ITO/PEN + G LSL and ITO/PEN + TG LSL were prepared by the drop-casting 5 μL solution containing 1 mg/mL GQD onto ITO/PEN and photoanode + T LSL. The highest PCE shown the photoanode + TG LSL (5.18%). The paper [68] presents the possibility of preparing flexible cells with a PEN substrate, which at standard illumination (100 mW/cm^2^) reached efficiencies of 2.6%, while at a lower intensity (18 mW/cm^2^) the efficiency of the devices increased to a value of 3.3%. Fu et al. [69] prepared flexible solar cells and tested the influence of platinum and polymer counter-electrode. The substrate on which the dye was anchored was ITO on PET covered with a TiO_2_ layer. The use of a polymer electrode resulted in an increase in V_oc_ and J_sc_ but higher resistances due to a decrease in FF. Finally, the Pt counter-electrode showed an efficiency of 2.95%, while the solar cell with polymer CE showed 2.18%.

### 2.2. Polymers in Mesoporous Layer of Photoanode

Currently, polymers are widely used to preparation of mesoporous oxide conductive layers to increase the porosity of material and thus the active surface area of the oxide. This chapter will briefly describe the polymers most commonly used to prepare the mesoporous metal oxide films. It is also worth noting that the increase in porosity improves yields a more effective penetration of the oxide substrate by the electrolyte, which will directly translate into an increase in short-circuit current. Additionally, and of particular importance, the increase in porosity will result in a larger number of dye molecules being able to anchor to the surface of the oxide semiconductor [70]. 

In order to create pores, polymers such as polystyrene, polyvinylpyrrolidone were used. In addition, copolymers are also used, which include P123 Pluronic (PEO_20_–PPO_70_–PEO_20_), copolymer of PVC and POEM. In [31], a P123 copolymer was used without and with addition of polystyrene with different particle sizes (62, 130 and 250 nm). A series of cells differing in oxide substrates were prepared. The highest efficiencies for the liquid electrolyte were obtained for the cell containing only copolymer P123 (1.58%). A cell prepared from a substrate containing 130 nm PS nanoparticles (1.44%) showed a slightly lower efficiency. However, cells containing polymer electrolytes also based on the iodine redox couple were prepared. In the case of the polymer electrolyte, the highest efficiency was achieved by cells with substrates containing P123 and PS-130 (0.89%) and the lowest device efficiency was determined for a solar cell with a substrate containing only copolymer Pluronic P123 (0.42%). The subject of [33] concerned the influence of the obtained oxide substrate using polymers on the photovoltaic performance of the cell containing the dye N719. In this study, a TiO_2_ paste was prepared using P123 to obtain a mesoporous layer, compared with a commercial P25 powder, and a TiO_2_ substrate containing a mixture of the two pastes was prepared. It was the use of a mixture of the two pastes that led to the highest yields of 6.50%, compared to the commercial one (4.00%). 

Park et al. [71] describe the comparison of mesoporous TiO_2_ layers prepared from a commercial paste containing P25 and a paste prepared using a PVC-g-POEM copolymer to obtain a higher degree of porosity and a larger active surface area. The N719 dye was used to prepare the devices, which showed an efficiency of 5.36% when using a commercial paste, while the PCE values increased to 7.45% when using a mesoporous TiO_2_ layer. A significant increase in the photocurrent density and fill factor was mainly observed. An important parameter from the point of view of developing the active surface is the number of adsorbed dye molecules. In the case of TiO_2_ mesoporous layer, a significant increase in dye loading values was observed in the conventional layer of 143 and 95.1 mmol cm^−2^, respectively.

### 2.3. Polymers as Dyes in DSSCs

Information on the use of polymers as dyes in DSSC-type cells is extremely rare in the literature. This can be seen in many review papers on the subject, in which no subsections on the subject appear. In preparing this work, a number of previous review papers were reviewed, where this information was omitted. This can be understood by the fact that finding this information in the literature is extremely difficult. However, it is worth noting that in the three cited papers, reasonably good yields were reported. Prakash and Subramanian [72] described the use of three poly(methacrylate)-based polymer sensitizers by employing phenothiazine (PPNPP), fluorine (FPNPP) and anthracene (APNPP). The anchor group for titanium oxide was NO_2_. The photovoltaic parameters were recorded for devices prepared with and without the addition of the co-adsorbent CDCA (chenodeoxycholic acid). The highest efficiency was shown by the cell in which PPNPP was used together with CDCA (4.12%). There were significant increases in the PCE values relative to both the other two compounds and the cell without the addition of CDCA. Ramasamy et al. [73] obtained three new poly-(methacrylate) bearing push–pull-type pendants oxindole-phenothiazine with tetrazole anchoring acceptor used as sensitizers in devices. High efficiencies of the prepared cells were observed, reaching as high as 5.91% for the POTZP3 compound. Wang et al. [74] described new conjugated polymers based on poly(triphenylamine-phenothiazine) with carboxylic acid side groups. Four compounds, including three polymers, designated as PAT, PPAT4, PPAT5 and PPAT6, respectively, were presented. Additionally, the PAT compound was identical to polymer PPAT4 and when comparing PCE the polymer achieved a higher value (4.7%). The highest of the compounds studied in this work. After discussing these selected works, it can be seen that although reports on the use of polymers as dyes in DSSCs cells are few, their performance is promising.

### 2.4. Polymers in Gel Electrolyte to Quasi-Solid State DSSCs

Liquid electrolytes containing triiodide/iodide or Co^2+^/Co^3+^ redox couples are now very commonly used. The use of liquid electrolytes, especially on a laboratory scale, is much simpler, quicker and cheaper than preparing a device with a solid electrolyte and applying a gold counter-electrode. However, the significant limitations of liquid electrolytes should be considered, such as volume change with temperature change, which significantly limits the operating conditions of the device. At high temperatures, the volume of the electrolyte increases considerably and evaporates, which makes it difficult to seal the solar cell; at low temperatures, the volume of the electrolyte decreases, which results in a decrease in the contact area between the electrodes and a decrease in the efficiency of the device. In addition, iodide electrolytes absorb part of the sunlight—the relatively low redox potential limits the available open-circuit voltage and causes corrosion of certain metals, making them unsuitable for use in a device such as silver [9,75,76]. Due to these limitations, solutions are being intensively sought to obtain similar PCE values to liquid electrolytes that also limit their sensitivity to changes in atmospheric conditions. For this reason, gel electrolytes, which obtain their consistency through the addition of polymers such as PEO, PEG, PAN or polyethylene glycol dimethyl ether (PGEDME), are now widely studied. All photovoltaic parameters of quasi-solid-state DSSCs based on the commercial Ru-dye are shown in Table 2.

In [34], the optimisation process of preparation and composition of the gel electrolyte was carried out to improve the performance of the device. The electrolyte contained triiodide/iodide redox couple, while the gel structure was obtained by using PEO and PGEDME. The study started with the choice of solvent, then focused on the amount of iodine in the electrolyte, ending with the addition of GuSCN. The study concluded that the most advantageous was the use of 0.2 M iodine and 0.1 M GuSCN which allowed to obtain V_oc_ = 793 mV, J_sc_ = 12.56 mA/cm^2^, FF = 0.77 and PCE = 7.66%. Furthermore, it should be noted that the final cell efficiency obtained was slightly lower than when using liquid electrolyte (8.03%). 

Huang et al. [35] prepared a DSSCs containing a PAN-based polymer electrolyte with iodine vapour redox, which was then doped with an azobenzene core compound. After the determination of photovoltaic parameters, a significant increase in the photocurrent density value and in the fill factor was found, thus changing the PCE from 4.19 to 6.28%. In addition, when compared to the liquid electrolyte, a negligible difference of 0.02% in favour of the liquid electrolyte was obtained. The influence of the effect of different amounts of copolymer on the photovoltaic performance obtained was studied by D. Kumar Shah et al. [36]. It was found that the addition of 7 wt.% PAN-co-PBA was the most beneficial, which caused a significant increase in both V_oc_ (646 mV), J_sc_ (13.16 mA/cm^2^), and PCE (5.23%). The study focuses on determining the optimum concentration of poly(vinylidene fluoride-co-hexafluoropropylene) to prepare a polymer electrolyte. Concentrations of 7–10 wt.% were investigated. It was found that the most favourable performance was obtained for a device with a concentration of 9 wt.%, for which the efficiency was 6.02%. 

The study described in [78] was devoted to the preparation of a polymer used to prepare a gel electrolyte. For this purpose, poly-3-(9H-carbazol-9-yl)propylmethacrylate (pCMA) was synthesised and then applied to a liquid electrolyte containing I^−^/I_3_^−^ redox couple. A commercial polymer polyvinylpyrrolidone (PVP) was used for comparison and was also used to prepare the electrolyte. For the new electrolyte, the lower efficiencies were obtained than for the commercial polymer, 2.20 and 3.00%, respectively, while it is worth noting that the current density increased significantly from 6.67 (PVP) to 10.30 mA/cm^2^ (pCMA). In another study [79], the effect of using a gel electrolyte obtained by adding prepared polyvinylidene fluoride (PVDF) or polydopamine- polyvinylidene fluoride (PDA@PVDF) to a commercial liquid electrolyte was studied. The polymer additives caused a decrease in the performance of the devices described by about one percentage point, which was mainly manifested by a decrease in the value of the short-circuit current density. Additionally, cells containing a gel electrolyte have been shown to have higher stability and less degradation over time. The work [80] concerned the preparation of gel electrolytes based on the electrospun polymer nanofibres (esPME) and composites (esCPME). PVdF-HFP fibres and PVdF-HFP composites with polypyrrole were obtained in the course of the research. The addition of polypyrrole resulted in an increase in cell efficiency of more than 7%. There was a slight improvement in the short-circuit current density (an increase of 0.8 mA/cm^2^). In work [81], the effect of the addition of polylactic acid (PLA) was investigated when metal-free dye MK-2 was used in the solar cell. A significant increase in the density of the generated short-circuit current relative to the liquid electrolyte (10.2 and 2.7 mA/cm^2^, respectively) was observed with PLA addition. This translated directly into an increase in device efficiency from 1.29 to 5.64%.

### 2.5. Conductive Polymers as Counter-Electrodes

The discovery of electrically conducting polymers has led scientists to delve deeper into a quest for replacing metals with their organic counterparts. Much progress has been made to replace the platinum metal counter-electrodes with organic polymers [82,83,84]. Counter-electrode being an indispensable component of DSSCs, catalyses the reduction of $I_{3}{}^{-}$ to $I^{-}$ by injecting electrons into the electrolyte and thus directly affects the device performance. Platinum metal is found to be the most suitable for this purpose due to its high conductivity and excellent electrocatalytic properties [85]. However, corrosion of the platinum, together with its high cost, makes it a less favoured choice for the counter-electrode, as it slowly deteriorates the stability of the device [86,87]. Use of expensive platinum metal as counter-electrode makes the device uneconomical. Hence, there is a need to replace the platinum metal electrodes with the noble-metal-free, low-cost, non-dissolving, electrically conducting and thermally stable materials that can alleviate the problems associated with platinum-based counter-electrodes.

Work continues in this field to develop such materials which can efficiently replace the metals-based electrodes. These include carbon materials [88,89], inorganic metal sulfides and oxides [90,91], conducting polymers [84,92,93,94], transition metal carbides and nitrides [95], alloys [95,96,97] and nanocomposites. The basic characteristics for a material to be used as counter-electrode in DSSCs include the optimum thickness of the active material, high electrocatalytic activity, porous structure, high surface area, good adhesion to substrate, and resistance against the corrosive electrolyte [98]. Among all other materials, conducting polymers have received special attention due to their low cost, facile synthesis and high electrical conductivity, along with other tunable properties. Here, we represent the most recent advances in the development of polymers for replacing metals as counter-electrodes.

#### 2.5.1. Polypyrrole as Counter-Electrode

The high electrical conductivity and catalytic properties authenticate polypyrrole as a considerable material for fabricating counter-electrodes for DSSCs. Various attempts have been made to exploit the efficiency of polypyrrole as a counter-electrode. The ease of synthesis and versatility in its high-yielding synthetic routes, which include both chemical and electrochemical polymerization or vapor phase oxidation, allows researchers to get attracted to polypyrrole [99]. Sangiorgi et al. exploited novel molecularly imprinted polypyrrole as counter-electrodes for dye-sensitized solar cells. This molecular imprinting approach allowed them to enhance not only the catalytic property of polypyrrole, but the selectivity of the catalyst was also increased. They improved the power conversion efficiency up to 20% [100]. Khan et al. synthesized porous polypyrrole by a simple hydrothermal method and employed it as a counter-electrode in DSSC. By mixing a minute quantity of copper perchlorate and a zeolitic-imidazole framework in porous polypyrrole, they received the power conversion efficiency of 8.63% and 9.05%, respectively [37]. In the most recent work, Saberi Motlagh et al. reported the fabrication of novel counter-electrode from polypyyrole-coated on carbon fabric. Electro-polymerisation was carried out to synthesize polypyrrole-coated carbon fabric. They achieved a power conversion efficiency of 3.86% [101]. However, much work is going on using polypyrrole as counter-electrode and to achieve the results which can enable us to replace platinum electrodes with polypyrrole-based counter-electrodes.

#### 2.5.2. Counter-Electrodes Based on Polypyrrole Nanocomposites

To improve the desired properties of polypyrrole, certain fillers are added to it or it is blended with appropriate materials. Various ways have been adopted to improve the required properties of polypyrrole so that it can be more efficiently used as a counter-electrode in DSSCs. Polypyrrole-covered graphene-based nanoplatelets are synthesized via electrochemical synthesis by Ohtani et al. The solar to electric power conversion efficiency of 4.30% was obtained, which is comparable to that of Pt-based counter-electrode (7.80%). η of PPy/GN-60 s containing DSSC was 3.3% which is even larger than DSSCs with Pt-based counter-electrode (3.00%) [102]. Another attempt to improve the performance of polypyrrole-based electrodes was done by Wu et al. Hybrid films from polyoxometalate doped polypyyrrole were prepared by electrochemical method and were exploited as counter-electrode in DSSCs. On average, the power conversion efficiency of these materials came out to be 6.19% [38]. Ahmed et al. incorporated polypyrrole with SrTiO_3_ nanocubes via oxidative polymerization method. Scanning through the various concentrations of strontium titanate, power conversion efficiency of 2.52% was obtained for 50 percent loading of SrTiO_3_. The incorporation of these particles into polypyrrole improved the efficiency of DSSCs from 1.29% by enhancing the surface area, electroactive response and catalytic property of the polypyrrole [103]. Rafique et al. reported the performance of DSSCs to 7.1% by synthesizing Cu-PPy-FWCNTs nanocomposites via dual step electrodeposition [104]. Relatively low conductivity and high charge-transfer resistance are two hurdles to paving the way for polypyrrole-based materials as counter-electrodes in DSSCs. The photovoltaic parameters of this type of solar cell are presented below in Table 3.

#### 2.5.3. Counter-Electrodes Based on Polyaniline

Intrinsically conducting polymers conceal potential members like polyaniline. In addition to polypyrrole, polyaniline has been widely used for the past three decades. Again, its facile synthesis, easy processability and tunable electric and redox properties make polyanilines unavoidable. Such properties are the result of three forms of polyaniline, viz. Leucomeraldine base (full reduced), Emeraldine base (half reduced form) and Pernigraniline base (full oxidized). Hence much of the research has been focused on fabricating counter-electrodes based on polyaniline.

Utami et al. synthesized nanostructured polyaniline using polyglyceryl-2-Dipolyhydroxystearate, a non-ionic surfactant. They used the nanostructured polyaniline to fabricate the counter-electrode for DSSCs. Through scanning of the surfactant concentration, they achieved the power conversion efficiency of 1.71% at 6% loading [106]. In another attempt, Krakus et al. exploited the polyaniline as the counter-electrode. Instead of the usual liquid electrolyte, they used cationic and anionic polymers as a quasi-solid electrolyte. The cationic quasi-gel electrolyte showed a higher electrolyte holding capability, elasticity and higher conductivity while the anionic one showed higher PCE. They were succeeded in achieving a high power conversion efficiency of 6.30% [39]. Furthermore, Jiao et al. studied the cyclic voltametric behavior of polyaniline by growing a thin film of PANI on a plastic substrate. By employing these thin films as a counter-electrode in DSSC, they gained a power conversion efficiency of 7.27% [107]. However, there is a need to improve the performance of PANI-based counter-electrodes. Scientists are attempting this by adding fillers in polyaniline substrates.

#### 2.5.4. Counter-Electrodes Based on PANI-Nanocomposites

Nano-sized fillers are usually added to polyaniline in order to improve both electrochemical and catalytic properties of PANI. The resulting materials usually show better properties than PANI itself. Here the most recent works on polyaniline nanocomposites for counter-electrodes in DSSCs are represented and all obtained photovoltaic parameters are gathered in Table 4. 

Farooq et al. fabricated four different counter-electrodes using four novel polyaniline-based materials. They fabricated counter-electrodes using polyaniline, ammonium lauryl sulfate doped polyaniline, sulphuric acid doped polyaniline and binary doped polyaniline containing both components. They achieved a power conversion efficiency of 4.54% [40]. Zatirostami et al. prepared tungsten oxide containing polyaniline nanocomposites and used them to fabricate counter-electrodes for DSSC. The nanocomposites bore good electrocatalytic behavior and high electrical conductivity. They succeeded in achieving a 12.8% improved power conversion efficiency of 6.78% as compared to Pt-based counter-electrode DSSCs [108]. In the most recent work published, Ravichandran et al. fabricated the counter-electrodes from FeS_2_ and achieved high FF value, higher Jsc and excellent electrocatalytic behavior towards electrolytes [109].

#### 2.5.5. Counter-Electrodes Based on Poly(3,4-Ethylenedioxythiophene)

After polyaniline, another most widely used polymer in DSSCs as their counter-electrode is Poly(3,4-ethylenedioxythiophene), abbreviated as PEDOT. Bella et al. synthesized ammonium ion-bearing poly(3,4-ethylenedioxythiophene), which they used as counter-electrode in DSSC. Even without using any expensive rare earth metal, they succeeded in achieving the power conversion efficiency of 7.02% [110]. In another attempt, Pradhan et al. fabricated DSSC counter-electrodes from poly(3,4-ethylenedioxythiophene) films and evaluated the performance of PEDOT by the varying film thickness. They prepared the PEDOT films having 33 nm, 65 nm and 120 nm thickness. Studies showed that the film with a 33 nm thickness produced the highest power conversion efficiency of 10.39%, while conversion efficiency of 8.11% and 7.45% were used by films of 65 nm and 120 nm thicknesses, respectively. The results clearly indicated that the performance declined with an incline in film thickness [111]. In a most recent work presented by Venkatesan et al. the optimum poly(3,4-ethylenedioxythiophene) film thickness was found to be 90 nm. They studied the DSSCs performance using PEDOT counter-electrodes under different light intensities. As the light intensity decreased, the efficiency of the cells tested increased. Under standard 100 mW illumination, the PCE of the cell was about 2.5% [112].

#### 2.5.6. Counter-Electrode Based on PEDOT-Nanocomposites

For obtaining more productive results using the PEDOT-counter-electrode, nanofillers are now added to the polymer substrate. Recently much work has been done to investigate the required catalytic and electrochemical properties of poly(3,4-ethylenedioxythiophene) nanocomposites. Mazloum-Ardakani et al. fabricated the DSSC counter-electrode from PEDOT-Ag/CuO nanocomposites. The cell showed the combined effect of poly(3,4-ethylenedioxythiophene), graphite and copper particles, thus obtaining an energy conversion efficiency of 9.06% [41]. Gaining confidence from the acquired results and in order to decrease the required amount of expensive platinum, Xu et al. fabricated counter-electrodes from transparent PEDOT film incorporated with Pt-nanoclusters. Enhancement in the electrochemical and catalytic properties in addition to improvement in film coverage, was observed. They prepared various counter-electrodes by varying the amounts of Pt-nanoclusters. The optimum concentration with the highest PCE was found to be in the Pt-10/PEDOT counter-electrode. By using the Pt-10/PEDOT electrode, the energy conversion efficiency was 6.77% when illuminating from the front side [113]. In a recent work by Gemeiner et al., various counter-electrodes were fabricated by screen-printing the poly(3,4-ethylenedioxythiophene): poly(styrene sulphonate). They doped the PEDOT:PSS with dimethyl sulfoxide, polyethylene glycol and ethylene glycol. Through optimizations, DSSC with a counter-electrode fabricated from PEDOT:PSS doped with 6 wt.% ethylene glycol showed a maximum power-converting efficiency of 3.12% [114].

Summing up the role of polymers in DSSCs, one can notice that polymers can be used in different forms and roles: as a flexible substrate, in the mesoporous layer of photoanode, dyes, in the gel electrolyte and as a counter-electrode.

In addition to the FTO-coated glass substrates, DSSC cells are increasingly being prepared on flexible polymer substrates made of PEN and also PET, which are coated with ITO. Considering another layer that forms the DSSC device, namely the conductive oxide layer, polymers are considered as materials used to receive the pores that develop the oxide surface. The most commonly used compounds are PS, PVP, as well as copolymers such as P123 Pluronic (PEO_20_–PPO_70_–PEO_20_) and a copolymer of PVC and POEM.

This review presents only a few latest papers on the use of polymers as dyes in DSSCs, but it is worth noting that most papers of this type do not mention this at all, and looking at the cases cited, polymers may be materials worthy of attention in this aspect as well. The use of polymers containing either a phenothiazine derivative in the main or side chain gives very good results. The use of polymers as additives in electrolytes to obtain gel structures and improve their time stability is being explored far more extensively than for polymeric dyes. PEO, PAN, and copolymers such as PAN-co-PBA, pCMA-PGE, or PVP-PGE are the most commonly used. However, it is not possible to compare parameter values in this case and to indicate the best polymer due to the use of different dyes and preparation methods.

Excellent electronic and catalytic properties are crucial for a material to be used in DSSCs as counter-electrodes. Expensive and corrosive platinum-based counter-electrodes are now being replaced by low-cost, non-corrosive, highly efficient and easily synthesized organic counter-electrodes. Polypyrrole, Polyaniline, Poly(3,4-ethylenedioxythiophene) and Poly(3,4-propylenedioxythiophene) are the main candidates in this regard. A blend of Poly(3,4-ethylenedioxythiophene) and Polystyrene sulfonate (PEDOT:PSS) is proved to be efficient enough to be replaced by Platinum-based counter-electrode. PEDOT:PSS is the most catalytic counter-electrodes among all polymeric counter-electrodes. Still, most of the research is focused on the development of more efficient carbonaceous material, which not only reduces the production cost of the solar cell but also increases the photovoltaic parameters along with its long-term stability. 

Different fabrication methods are being employed to synthesize materials for counter-electrodes with optimum required properties. Porous or network structured polymer material with high thickness can give better electrocatalytic properties. Electrochemical polymerization is an ideal technique to obtain polymeric materials with desired properties and dimensions. However, this technique has some limitations when applied on a large scale. Moreover, nanocomposites of these polymers are synthesized to further improve the available catalytic surface area and electronic properties of the material. For instance, depositing uniform films is a challenging task for PANI-based counter-electrodes. Hence, finding appropriate nanofillers improves the conductivity but also increases the surface area by acting as pore former without disturbing the film uniformity. The advantage of Polypyrrole-based counter-electrodes is that by carefully selecting the fabrication method and dopants, one can simultaneously obtain the benefits of high-catalytic and energy-storing properties of the polypyrrole. 

Moreover, from the above-presented literature review of polymers-based counter-electrodes for DSSCs, we conclude that the conductive polymers with excellent electrochemical and catalytic properties can selectively catalyze the redox reaction of the electrolyte and can further improve the photovoltaic parameters of the solar cells. 

## 3. Polymers in Bulk-Heterojunction Solar Cells (BHJ)

BHJ solar cells are characterised by their layered structure, but it is important to note that the active layer is a donor and acceptor blend. The simplest architecture of both conventional and inverted systems is shown in Figure 2.

### 3.1. Polymers as Donors Materials

As for other types of solar cells, the power conversion efficiency is the most important parameter to assess the efficiency of a fabricated solar cell, which in turn depends directly upon short-circuit current density, open-circuit voltage and fill factor [42]. This fact prompts the researchers to look deeply into the factors which directly impact these parameters. Thus, plenty of work is being done to improve the PCE of bulk heterojunction solar cells by fine-tuning the (1) HOMO of donors to the deep-lying level and LUMO of the acceptor to low lying level for the improvement of V_oc_, (2) incorporation of strong acceptors in D-A kind of donor systems, which alleviates the intramolecular charge transfers, (3) photon harvesting capability which in turn improves the external quantum efficiency and incident photon to current conversion efficiency, (4) use of superior charge carrying polymers as Donors and Acceptors to avoid energy loss due to charge recombination and (5) solution processability by increasing blending ability [43]. Out of these parameters, J_sc_ and V_oc_ are directly affected by the nature of the donor material. Hole-transporting ability and HOMO levels of donor directly impact the current density and open-circuit voltage, respectively. Hence it is inevitable to engineer the donor molecules that can efficiently improve the PCE of BHJ solar cells. 

The main roles of the donor material are to absorb sunlight and transport the holes to the respective electrode. Certain requirements for organic semiconductors for employing them as donor materials in BHJ must be kept in mind. The first step is the exciton generation by absorbing solar light and most of the fraction of sunlight is absorbed by the donor material; therefore, it is necessary to optimize the absorption range of the donor materials to efficiently cover most of the solar spectrum. The use of low-bandgap donor materials, i.e., materials with a bandgap lower than 2 eV, is the key to this arduous task [44,115]. For instance, donor material having 1.1 eV bandgap can efficiently cover 77% part of AM 1.5 solar photon flux, while 1.9 eV bandgap donor material can cover only 30% of it [115]. Effective light absorption is achieved by employing π-conjugated polymers surely because of their superior absorption coefficient, i.e., 10^7^ m^−1^ [116]. The thickness of the active layer in BHJ solar cells must be kept small (100 nm) so as to allow maximum exciton diffusion. It is due to the low charge carrier mobility of the polymers. The low thickness of the active layer makes the BHJ more cost-effective compared to the inorganic silicon-based solar cells, where the thickness of several micrometres is required. After exciton generation and their diffusion, the next step is the dissociation of excitons into separate charged particles, i.e., holes and electrons. The frontier orbital energy levels of the donor and acceptor molecules decide the efficiency of excitons dissociation [117,118]. The energy offset between the Acceptor LUMO and Donor LUMO should be between 0.1–1.4 eV for the dissociation of exciton into electrons and holes [119]. Herein, we review the most recent advances in the polymer donors for BHJ solar cells.

#### 3.1.1. Wide-Bandgap Polymer Donors

The narrow absorption range and lower charge carrying ability had limited the wide applicability of poly(2-methoxy-5-(2′-ethyl-hexyloxy)-1,4-phenylene vinylene), which was the first ever polymer donor for organic solar cells [120]. Since then, a copious amount of research has been devoted to synthesizing an appropriate polymeric donor material having a wide range of spectral absorption by fine-tuning the bandgap. The PV parameters of the described devices are shown in Table 5.

The chemical structures of polymers used as wide bandgap materials are shown in Figure 3.

Wang et al. synthesized a 1,2-difluoro-4,5-bis(octyloxy)benzene wide-bandgap (2.16 eV) polymer W1 and used it with Y6 (BTP-4F non-fulerene electron acceptor) to fabricate a BHJ solar cell in the architecture of ITO/PEDOT:PSS/W1:Y6/PDIN/Ag. They succeeded in obtaining a PCE as high as 16.16% [121]. Cao et al. synthesized a wide-bandgap polymer PBDT-TTz based on thiazolothiazole motif and exploited the role of the imine substitution on the electronic charge transport and optical properties of the polymer. For comparison, a non-imine polymer PBDT-TT was also synthesized, and it was found that the imine substitution not only improved donor-acceptor miscibility, but also increased the face-on orientation and crystallinity of the donor phase. ITO/PEDOT:PSS/PBDT-TTZ:N2200/PFN-Br/Ag architecture was fabricated to measure the photovoltaic parameters. An imine-substituted polymer donor exhibited a power conversion efficiency of 8.4% as compared to non-imine polymers, which showed a 0.7% PCE value [122]. Changguo et al. synthesized two polymers based on 4,8-bis(5-(2-ethylhexyl)thiophen-2-yl)benzo[1,2-b:4,5-b′]dithiophen, and a combination of thiazolothiazole and thiazoles. Donor polymers PTzTz and PBTz, when used along with a non-fluorine Acceptor (IT-4F) in the architecture of ITO.PEDOT:PSS/PTzTz:IT-4F/PFN-Br/Al and ITO.PEDOT:PSS/PBTz:IT-4F/PFN-Br/Al delivered energy conversion efficiencies of 10.63% and 8.76%, respectively [123]. Another wide-bandgap (1.98 eV) donor copolymer, as shown in Figure 3, was synthesized and employed in BHJ-solar cells by Qishi et al. By fabricating the cell in the ITO/PEDOT:PSS/D18:Y6/PDIN/Ag architecture provided a final PCE of 18.22% [15].

In a separate attempt, Jianqiang et al. fabricated a thick active layer of BHJ solar cells by adding PCBM into a D18-Y6 blend and used the architecture ITO/PEDOT:PSS/D18:Y6:PC61BM/PDIN/Ag. Increasing the thickness to 110 nm, they achieved a power conversion efficiency of 17.89% [124]. Advancing further, Zhao et al. reported bithieno[3,4-c]pyrrole-4,6-dione (PBiTPD), a donor based on the thieno[3,4-c]pyrrole-4,6-dione (TPD) motif. Using the solar cell structure ITO/PEDOT:PSS/PBiTPD:Y6/PFN-Br/Ag, the power conversion efficiency was 14.2% [125]. Keshtov et al. fabricated the binary and ternary BHJ solar cells by employing a D-A polymer P106 as a donor and two non-fluorine acceptors, Y18-DMO and DBTBT-IC. P106 contained 2-dodecylbenzo[1,2-b:3,4-b′:6,5-b″]trithiophene (3TB) as a donor unit with dithieno [2,3-e;3′2′-g]isoindole-7,9 (8H) (DTID) as an acceptor unit. Two binary solar cells having the architectures of ITO/PEDOT:PSS/P106:DBTBT-IC/PFN/Al and ITO/PEDOT:PSS/P106:Y18-DMO/PFN/Al produced power conversion efficiencies of 11.76% and 14.07%, respectively, while the ternary solar cell with the structure ITO/PEDOT:PSS/P106: DBTBT-IC:Y18-DMO/PFN/Al obtained a power conversion efficiency of 16.49% [126]. In a more recent work on wide-bandgap donor polymers, Gokulnath et al. reported the fabrication of a ternary solar cell based on the siloxane-functionalized polymer Si-BDT. The ITO/ZnO/[PTB7-Th(0.6):Si-BDT(0.4):DCNBT-TPIC(0.6)/MoO_3_/Ag architecture provided a power conversion efficiency of 13.45% [127].

#### 3.1.2. Medium-Bandgap Polymer Donors

Thieno[3,4-b]thiophene or benzodithiohene-based polymers have produced satisfactory results when employed as donor materials in BHJ solar cells [128,129]. The photovoltaic parameters of the described devices are collected in Table 6 and the structures of the polymers are shown in Figure 4.

Chen et al. described the enhanced photovoltaic performance by using a novel medium bandgap polymer as the donor for BHJ. A copolymer based on thiophene, diflurobenzothiadiazole (FBT) and benzodithiophene (BDT) motifs were synthesized and employed to fabricate ternary and binary BHJ solar cells. Non fluorine 3,9-bis(2-methylene-(3-(1,1-dicyanomethylene)-indanone))-5,5,11,11-tetrakis(4-hexylphenyl)-dithienol[2,3-d:2′,3′-d′]-s-indaceno[1,2-b:5,6-b′]dithiophene (ITIC) and [6,6]-phenyl-C71-butyric acid methyl ester (PC71BM) were used as Acceptors.

The solar cells were fabricated in the following architectures: ITO/PEDOT:PSS/POBDFBT:ITIC/PFN/Al, ITO/PEDOT:PSS/POBDFBT:PCBM/PFN/Al and ITO/PEDOT:PSS/POBDFBT:PCBM:ITIC/PFN/Al. Power conversion efficiencies were calculated to be 6.16%, 6.23% and 7.91%, respectively [130]. Sharma et al. fabricated a ternary solar cell using BODIPY-thiophene-based conjugated polymer. The ternary cell was fabricated by mixing two polymers with two different acceptors ITIC-m and Y6, in a weight ratio of 1:0.3:1.2.

Fabricating architecture of the ternary solar cell was ITO/PEDOT:PSS/P:ITIC-m:Y6/PFN/Al and it delivered a power conversion efficiency of 15.13%, which is higher as compared to the binary solar cells, i.e., 12.10% for P-ITIC-m (1:1.5) and 13.16% for P-Y6 (1:1.5) [131]. An and Qiaoshi et al. succeeded in achieving a PCE of 17.22 percent by employing ternary strategy of fabricating solar cells. For this PM6, a donor polymer was used along with Y6 and MF1 as acceptors in the architecture of ITO/PEDOT:PSS/active layer/PDIN/Al [132]. Separately, Yuan and Jun et al. used a ladder-type Y6 as acceptor and PM6 as a donor and succeeded in achieving a PCE of 15% [133]. Yan et al. fabricated ternary solar cells by incorporating PCBM as a third component in the PM6-Y6 binary mixture. The architecture used by them was ITO/PEDOT:PSS/PM6:Y6 (w and w/oPC_71_BM)/PDINO/Al and they obtained the PCE values of 16.67% in rigid and 14.06% in flexible organic solar cells [134].

Another attempt to successfully employ medium bandgap polymers as donors in BHJ devices was made by Penget et al. They synthesized a D-A type polymer PM7-Si after modifying well-known PM6 by replacing the ethylhexyl group with alkylsilyl chains and fluorine atom with chorine. The fabrication of the ternary BHJ solar cell was made with a structure of ITO/PEDOT:PSS/PM6:PM7-Si:C9/PFN-Br/Ag. The obtained PCE of the ternary cell was 17.7% which was higher as compared to a binary cell based on PM6:C9 [135]. In a more recent work, a medium bandgap copolymer donor D-A_1_-D-A_2_ was synthesized where D is thiophene, A_2_ is novel anthra[1,2-b:4,3,b′:6,7-c″]trithiophene-8.12-dione (A3T) and A_1_ is fluorinated benzothiadiazole in case of P130 or benzothiadiazole in case of P131. The architecture of the cell was ITO/PEDOT:PSS/P130 or P131:Y6/PFN/Al. The power conversion efficiencies are 15.28% and 11.13% for P130 and P131, respectively [136].

#### 3.1.3. Narrow Bandgap Polymers

Researchers are doing continuous work on optimizing the bandgap width of the polymers to employ them efficiently as the donor materials in BHJ solar cells. The PV parameters of described devices are collected in Table 7.

Pan et al. reported diketopyrrolopyrrole (DPP)-based polymer PffBT-DPP has a narrow bandgap of 1.33 eV. One non-fullerene acceptor MeIC and one fullerene acceptor PCBM were employed to fabricate both binary and ternary solar cells with the architecture ITO/PEDOT:PSS/active layer/ZrAcAc/Al. The ternary device PffBT-DPP:PCBM:MeIC showed a power conversion efficiency of 9.0% while binary BHJ solar cells showed PCE of 6.8% and 2.0% for PffBT-DPP:PCBM and PffBT-DPP:MeIC respectively [137]. A PT10-based donor polymeric material was synthesized by Rech et al. and was used in the architecture of (ITO)/poly(3,4-ethylenedioxythiophene):polystyrenesulfonat(PEDOT:PSS)/PTQ10:Y6/PFN-Br/aluminum. They acquired the PCE of 15%. In [138], Caliskan et al. synthesized a donor material based on benzo dithiophene by attaching a 2-(2-octyldodecyl)selenophene ring at the fourth and eighth position of benzene ring in BDT. The structure of the solar cell was ITO/PEDOT:PSS/Polymer:PC_71_BM/LiF/Al and the obtained PCEs were 2.36%, 2.07% and 2.45% for P1, P2 and P3, respectively [139]. Guo et al. synthesized narrow bandgap (1.6 eV) conjugated polymers based on bis(2-alkyl)-5,8-dibromo-6,7-difluoroquinoxaline-2,3-dicarboxylate (EF-Qx) unit. For better performance D-A (PTT-EFQX) and D-A_1_-D-A_2_(PT-DFBT-T-EFQX)-type materials were synthesized in the architecture of ITO/PEDOT:PSS/PTT-EFQX:PCBM/PFN-Br/Ag and ITO/PEDOT:PSS/PT-DFBT-T-EFQX:PCBM/PFN-Br/Ag respectively. PCE of the solar cells containing D-A type structure was found to be 5.37%, while D-A_1_-D-A_2_ type has 2.69% [140]. Can et al. recently synthesized low bandgap (1.30–1.35 eV) D-A copolymers 4,4-Difluoro-4-bora-3a,4a-diaza-s-indacene (BODIPY) as donor part and benzo[1,2-b:4,5-b′]dithiophene (BDT) acting as acceptor. The highest conversion efficiency of 4.40% was shown by P(T2BDY−TBDT), having a very high current density of 12.07 mAcm^−2^ [141].

### 3.2. Polymers as Acceptor Materials

To a lesser extent, polymers are used as acceptors in BHJ cells. However, in the literature in recent years, few reports can be found on the use of polymers as acceptors (Table 8).

Zhu et al. [47] prepared and described a series of solar cells with structure ITO/PEDOT:PSS/PTzBISi:N2200/C60N/Ag prepared under different conditions. N2200 polymer was used as an acceptor. Two solvents, such as chlorobenzene and 2-methyltetrahydrofuran were used, in addition to the thermal annealing and solvent vapour annealing. It has been shown that 2-methyltetrahydrofuran and thermal annealing together with solvent vapour annealing are the most favourable applications. For the solar cells prepared in this way, high photovoltaic parameters of V_oc_ = 880 mV, J_sc_ = 17.62 mA/cm^2^, FF = 0.76 and PCE = 11.25% were obtained. The values given were averages and the maximum efficiency of the device was as high as 11.76%.

In [48], polymeric acceptors were used, which are naphthalene-diimide and perylenediimide derivatives. The prepared inverted structure devices showed high efficiencies of 8.59% (NDP-V) and 7.38 (PDI-V). When NDP-V was used, an increase in both Jsc and FF was observed (by 1.68 mA/cm^2^ and 0.03, respectively). Nagesh et al. [49] prepared a series of inverted photovoltaic cells containing a copolymer as an acceptor. The fabricated devices differed in the ratio of NDI-biselenophene/NDI-selenophene copolymer repeating units. As a result of the research carried out, it was found that the most advantageous was the use of an NDI-biselenophene/NDI-selenophene copolymer with an equivalent proportion of 90:10 (BSS10); for this acceptor structure, yields of over 10% were obtained. In [143], a block copolymer containing donor and acceptor moieties (PBDB-Tb-PYT) was used. The effect of the solvent used (chloroform, chlorobenzene, o-dichlorobenzene, tetrahydrofuran, toluene) was studied. Moreover, the addition of 1-chloronaphthalene was used. By using the CN additive, a significant increase in both Jsc and FF and therefore, in efficiency, was observed (from 7.18 to 11.32%). The obtained photovoltaic performance results for the copolymer active layer were compared with the blend obtained by mixing the donor (PBDB) and acceptor (PYT), respectively. A PCE of 14.57% was recorded for the blends obtained by mixing the polymers in the active layer.

On the basis of the above-presented results from the latest scientific reports, we can summarize the role of polymeric materials in BHJ solar cells. The most important light-harvesting responsibility of the donor material in bulk heterojunction solar cells compels the researchers to choose the optimum bandgap materials in this regard. Inorganic silicon-based solar cells require the thick active donor layer and hence not only increase the production cost but also are based on a non-renewable silicon source. Therefore easily synthesized, low-cost, environmentally friendly and thermally stable polymeric donor materials are continuously increasing in demand. Low-bandgap donor polymers are optimum for bulk heterojunction solar cells because they absorb most parts of the solar spectrum and are thus efficient light absorbers. Non-fullerene acceptors are more compatible with the polymer donors because of the lowered LUMO levels and high extinction coefficient.

The wide visible light absorption range is the specialty of the polymers only. Poly(3-hexylthiophene) (P3HT) is the most widely used polymer donor with PCBM acceptor. Benzo[1,2-b:4,5-b′]dithiophene (BDT)-based donor polymers are among the best polymers used against fullerene and non-fullerene acceptors. Not only binary, but ternary strategies are employed to further increase the efficiency of all-polymer solar cells. PCEs greater than 11% have been achieved by fullerene-based acceptors, while PCEs greater than 17% have been accomplished by non-fullerene BHJ solar cells. Regioregular geometry of the polymers controls the polymeric chain supramolecular assembly and thus influences the charge transport properties. Developing novel synthetic methodologies has become crucial for controlling regioregular geometry of the polymers during copolymerization. From this literature review, it can be noted that good miscibility between acceptor and donor is another important parameter, which must be kept in mind to achieve high-performing bulk heterojunction solar cells. Side-chain engineering plays an important role in the electron-donating abilities of the donor material. Therefore, it is the need of the hour to develop facile and cost-effective synthetic methods for synthesizing polymer donors with optimum properties. Polymers used as acceptors in BHJs are far less common, although there is, of course, information on this in the literature. The most commonly used polymeric acceptors are N2200, PBDB-T or NDP-V, but it is impossible to compare the PV performance of these devices due to different preparation methods and architecture.

## 4. Polymers in Perovskite Solar Cells (PSCs)

Over the last two decades, the significant development of the sourcing electricity concept from solar energy is observed. The current research topic in photovoltaics is perovskite solar cells. The PSCs are cells of the latest technology, for which has been noted a very fast increase in efficiency (PCE) from 3.8% in 2009 to 25.2% in 2020, which may indicate that this type of cell will find commercial applications [12,144,145]. The perovskite solar cells are a hybrid system, a combination of organic and inorganic structures. A perovskite can be represented by a general formula ABX_3_, where A is the organic ion (the most common is methylammonium ion −[CH_3_NH_3_]^+^), B is Pb^2+^ ion, Sn^2+^ or Cd^2+^, and X is a halogen ion I^−^, Br^−^ or Cl^−^. The perovskite is characterized by wide absorption of visible and near-infrared radiation, low binding energy exciton (~2 meV), and a direct bandgap. Additionally, perovskite materials show: (i) a long time carrier life (~270 ns), which generates the length of the diffusion path at the level of ~1 μm in thin layers and up to ~175 μm in single crystals, thus ensuring hassle-free transport of charge carriers through the absorber (perovskite) 300 nm thick (no recombination effect), (ii) high mobility load carrier (up to ~2320 cm^2^ V^−1^ s^−1^); and (iii) high dielectric constant (~18–70), which makes them ideal materials for photovoltaics [146]. A perovskite absorber, hole-transporting material (HTM), electron transporting material (ETM), and electrodes are all common components in PSC devices. Photo-generated electrons/holes in the perovskite absorber are transported to the ETM/HTM and selectively collected by the anode/cathode when a PSC is illuminated. Both n–i–p (traditional) and p–i–n (inverted) forms of PSCs can operate successfully due to perovskites’ ambipolar charge transport feature [147].

This work is a presentation of the current achievements concerning the applications of polymers in perovskite solar cells, which will be shown in the following subsections of this publication. The conventional and inverted structures of PSCs are presented in Figure 5a, while the chemical structures of polymers used in PSCs are shown in Figure 5b.

### 4.1. Polymers in Improving Perovskite Morphology

The large-scale development of perovskite solar cells requires high-quality failure-free perovskite foils with better surface coverage. Several solutions to this problem will be presented in this section.

Zhao et al. presented a polymerization-assisted grain growth (PAGG) technique for obtaining stable and efficient perovskite solar cells with FA_1−x_MA_x_PbI_3_. DI (Dimethyl Itaconate) monomers were added to the PbI_2_ precursor (1.0% molar ratio) to provide sufficient contact between the PbI_2_ and their carbonyl groups (sequentially deposited approach). An in situ polymerization process was started during the subsequent PbI_2_ annealing process, leaving the as-formed heavier polymers adhering to the grain boundaries with previously set contact. Due to the adequate polymer-PbI_2_ interaction, there was a higher energy barrier for producing perovskite crystals when reacting with FAI (formamidinium iodide), resulting in more sizeable crystal grains. Furthermore, the carbonyl groups of polymers were led to the under coordinated Pb^2+^ and effectively diminished the trap density, whereby a PCE of 23.0% was obtained. Effective passivation, combined with the hydrophobic character of the polymer, significantly slowed the rate of deterioration, resulting in significant increases in stability [148].

Furthermore, Yousif and Agbolaghi investigated the potential application of the rGO and CNT precursors and their derivatives grafted with the rGO-g-PDDT and CNT-g-PDDT (irregioregular) and CNT-g-P3HT and CNT-g-P3HT (regioregular) polymers to improve the morphological, optical, and photovoltaic properties of FTO/b-TiO_2_/mp-TiO_2_/CH_3_NH_3_PbI_3_/spiro-OMeTAD/MoO_3_/Ag perovskite devices (the ratio of carbonic materials to the perovskite was 1:15). The perovskite system behaviour (cell performance) was modified by the type of rGO or CNT (carbonic materials) and the regioregularity of grafts. The best results of PCE were obtained with CNT nanostructures grafted with P3HT backbones, which were 16.36% [149].

Yu et al. elaborated a new p-type p-conjugated ladder-like polymer P-Si (poly(3,30-(((2-(4,8-bis(5-(2-ethylhexyl)thiophen-2-yl)-6-methylbenzol[1,2-b:4,5-b0]dithiophen-2-yl)-5-methyl-1,4-phenylene)bis(oxy))bis(hexane-6,1-diyl))bis(1,1,1,3,5,5,5-heptamethyltrisiloxane)) for perovskite solar cells-based on SnO_2_. This introduced of a small amount of P-Si into an anti-solvent to improve the morphology and crystallinity of perovskite films. The P-Si (the HOMO energy level is −5.41 eV) could act as a hole-transport medium between the spiro-OMeTAD and the perovskite layer (enhanced hole transportation). As a result, the highest PCE of solar cell with P-Si (0.1 mg ml^−1^) was achieved at 21.3% [150].

Fu et al. also applied a polymer in the anti-solvent process to passivate the defects of perovskite films and dominate the perovskite crystallization. The researchers used C60-PEG (fullerene end-capped polyethylene glycol). The application of C60-PEG also influenced the surface of the perovskite films. As a consequence, the highest PCE (17.71%) of the tested perovskite solar cells was registered [151].

Moreover, Chen et al. exploited a PBTI (poly(bithiophene imide)) in the anti-solvent step of the perovskite deposition process, resulting in effective passivation of the grain boundary defects and thus improvement of the tested devices performance. The PBTI (0.25 M) may be efficiently incorporated into grain boundaries (grain boundary defect passivation) cause of a vast lower in recombination losses and the ensuing increase in V_oc_ and PCE (20.57%) [152].

Yao et al. demonstrated that a polymer alloy of a PS (polystyrene) and a PMMA (poly (methyl methacrylate)) could profit the crystal growth and boost the flexibility of the perovskite solar cells. The polymer alloy (AMS, PS:PMMA, i.e., 1:2) was integrated with the perovskite layer (CH_3_NH_3_PbI_3_) during the anti-solvent process. The additive of AMS may boost the grain size of perovskite crystals and suppress the crystallization of the absorber layer. As a result, the PSCs with AMS showed a PCE of 17.54% [153].

For the first time, Suwa et al. incorporated a small amount of a PTMA (poly(1-oxy-2,2,6,6-tetramethylpiperidin-4-yl methacrylate) into the perovskite layer, thereby increasing the durability of the perovskite. The superoxide anion radical generated following light irradiation on the layer was eliminated by PTMA, which could react with the perovskite molecule and degrade it into lead halide. The photovoltaic conversion efficiency of a cell made with a PTMA-incorporated perovskite layer (0.3 wt.% amount of the polymer vs. the perovskite) and a hole-transporting PTAA (polytriarylamine) layer was 18.8% [154].

Additionally, for the perovskite surface, Chen et al. used a PEA (poly(propylene glycol) bis(2-aminopropyl ether)) and applied grain boundary passivation. PEA’s unshared ether–oxygen electron pair activates, forming a crosslinking complex with lead ions, thus lowering the trap state density and inhibiting non-radiative recombination in perovskite films. The PCE of the MAPbI_3_-based cells with PEA was 18.87% (1 wt.% of PEA) [155].

In addition, Garai et al. designed and synthesized a PHIA (poly(p-phenylene)) as additives to the perovskite precursor solution. The side chains of the polymer were selectively functionalized, allowing it to be used in the effective trap passivation of perovskites. The PHIA polymer caused the production of perovskite films of a higher quality and with bigger grain sizes. The passivated device exhibited minimum charge collection at the interface, lower recombination and lesser traps, allowing for an improved charge transfer. As a result of the passivation, the device had a high PCE of 20.17% (0.50 mg mL^−1^ of PHIA) [156].

In contrast, Zarenezhad et al. utilized a PPy (polypyrrole) in the precursor solution to fabricate mixed halide devices. PPy was used as a conductive compound to ensure an enhanced electron-hole extraction and transfer. The PPy additive amended the layer quality by mitigating the growth of the perovskite crystals (the lower charge carrier recombination and efficient carrier extraction). The highest achieved PCE of perovskite solar cells (1 wt.% of PPy) was 13.2% [157].

Zhong et al. employed a mixture of a PVP (polyvinylpyrrolidone) and a PEG (polyethylene glycol) with an appropriate mass ratio in precursor solution to perfect the morphology of perovskite, optical and photovoltaic properties, and air stability of perovskite (CH_3_NH_3_PbI_3_) films. After modifying the perovskite film with a polymer mixture (PVP and PEG), the crystallinity, uniformity, smoothness, compactness, and surface coverage of the perovskite film improved. The air stability of the tested PSCs could be imputed to the unique hygroscopicity of the polymer mixture. The bondings between polymer mixture and perovskite also contributed to the inhibition of ion migration and the synergistic stabilization of the perovskite structure [158].

### 4.2. Polymers as Hole-Transporting Materials (HTM)

Hole-transporting materials are essential elements of perovskite cells. Compounds acting as HTMs in PSCs should be of: (i) an appropriate level HOMO (i.e., Highest Occupied Molecular Orbital, which allows the band’s energy valence perovskite material to be adjusted), (ii) high hole mobility, (iii) wide light absorption spectral range, (iv) photochemical stability, and (v) good layering ability [145,159]. The conjugated polymer HTMs have good stability and solution operability when compared to organic small molecule HTMs and inorganic HTMs. Photovoltaic parameters of PSCs with polymeric HTM are presented in Table 9.

Chawanpunyawat et al. developed an IDTB (poly(1,4-(2,5-bis((2-butyloctyloxy) phenylene)-2,7-(5,5,10,10-tetrakis(4-hexylphenyl)-5,10-dihydro-sindaceno[2,1-b:6,5-b′]dithiophene))) as dopant-free polymeric HTM. IDTB was shown a high mobility and an intensive interaction of the backbone to perovskites through IDTB’s S/O atoms (a high holeextracting ability) and also an effective passivation of the defects in absorber layer. The prepared PSCs with IDTB as dopant-free HTM were attained PCE (19.38%) comparable to the devices with doped spiro-OMeTAD (2,2′,7,7′-tetrakis[N,N-di(4-methoxyphenyl)amino]-9,9′-spirobifluorene) (18.22%) [160].

For the first time, Liao et al. applied a P3CT (poly[3-(4-carboxybutyl)thiophene-2,5-diyl]) as HTM in perovskite solar cells. The P3CT has demonstrated ideal dual functionality for device applications thanks to a plethora of carboxylic groups (−COOH) on the side chains. Molecules of the P3CT could firmly attach to the ITO surface, following it to achieve a work function that was similar to that of the perovskite active layer. To eliminate recombination defects, the Lewis base character of the −COOH group could efficiently passivate the under-coordinated Pb^2+^ ions at the HTL/perovskite interface. As a consequence of using the P3CT as HTL, a significant PCE of 21.09% was successfully produced [161].

Qi et al. proposed a PBT1-C obtained from the copolymerization between 1,3-bis(4-(2-ethylhexyl)thiophen-2-yl)-5,7-bis(2-alkyl)benzo[1,2-c:4,5-c′]dithiophene-4,8-dione units and benzodithiophene. The PBT1-C was able to passivate the surface traps of the perovskite layer and was characterized by excellent hole mobility. Through its carbonyl (−CO) functional groups, PBT1-C might passivate under coordinated defective perovskites, reducing nonradiative recombination and enhancing charge extraction. The tested PSCs was shown a PCE of 19.06% [162].

Jeong et al. reported a PCDTBT (poly[N-9′-heptadecanyl-2,7-carbazole-alt-5,5-(4′,7′-di-2-thienyl-2′,1′,3′-benzothiadiazole)]) as an efficient hole-transfer material (0.02 cm^2^ V^−1^ s^−1^). The greatest fracture energies in the perovskite devices were caused by PCDTBT fibrils produced at the grain boundaries of the perovskite layer. These energies have offered extrinsic reinforcement and shielding for improved mechanical and chemical stability. The PSCs with PCDTBT exhibited a PCE of 14.08% as well as significantly enhanced mechanical and air stability [163].

Yao et al. synthesized polymeric HTMs by inserting a phenanthrocarbazole unit into polymeric thiophene or selenophene chain (PC1, PC2, and PC3). The addition of a planar and broad phenanthrocarbazole unit was dramatically enhanced by the adjacent polymer strands’ π−π stacking and interactions with the perovskite’s surface. As a result, the PSC with PC3 as a dopant-free HTM had a stable PCE of 20.8% and a greatly increased lifetime [164].

Ma et al. explored a J71 (poly [[5,6-difluoro-2-(2-hexyldecyl)-2H-benzotriazole-4,7- diyl]-2,5-thiophenediyl [4,8-bis [5-(tripropylsilyl)-2-thienyl]benzo[1,2-b:4,5-b’]dithiophene-2,6-diyl]-2,5-thiophenediyl]), PBDB-T (poly[(2,6-(4,8-bis(5-(2-ethylhexyl)thiophen-2-yl)benzo [1,2-b:4,5- b0 ]dithiophene)-co-(1,3-di (5-thiophene-2-yl)-5,7-bis(2-ethylhexyl)-benzo[1,2-c:4,5-c0 ]dithiophene-4,8-dione)]), and PM6 (poly [(2,6-(4,8-bis(5-(2-ethylhexyl-3-fluoro) thiophen-2-yl)-benzo [1,2-b:4,5-b0 ]dithiophene))-alt-(5,5-(10,30 -di-2-thienyl-50,70 -bis(2-ethylhexyl)benzo [10,20-c:40,50 -c0 ]dithiophene-4,8-dione)]) in PSCs. The alignment of the perovskite and HTM energy levels were crucial for hole extraction and recombination suppression at the interface. The fundamental techniques for obtaining a high-performance device were to increase the material carrier conveying capacities while retaining low-charge recombination [172].

You et al. developed PBDTT and PBTTT (D-A polymers) high-efficiency HTMs of PSCs. In the PBDTT and PBTTT, IDT or IDTT was used as the D unit, BDD served as the A unit, and thienothiophene acted as a π-bridge. The n-i-p PSCs incorporating these polymer HTMs displayed a highly promising device performance (PCE of around 20%). The devices with PBDTT functioned slightly better than PBTTT because of the superior solubility of IDT, which resulted in a smoother film and better perovskite/HTM/anode interfacial contact [165].

Shalan et al. investigated a new polymeric HTMs P(mPhDTP) (poly(1-(4-methoxyphenyl)-2,5-bis(5-methylthiophen-2-yl)-1H-pyrrole)), P(hPhDTP) (poly(1-(4-hexylphenyl)-2,5-bis(5-methylthiophen-2-yl)-1H-pyrrole)), P(hBT) (poly(3-hexyl-5,5′-dimethyl-2,3′-bithiophene)) and P(BT) (poly(5,5′-dimethyl-2,3′-bithiophene). These obtained polymers were discovered to be extremely soluble in a variety of halogenated and non-halogenated solvents, making them eco-friendly materials. The HOMO/LUMO band positions of the tested HTMs were aligned with those of perovskite, guaranteeing that holes were extracted from the CH_3_NH_3_PbI_3_ to the HTM layer with appropriate driving force. The highest PCE (15.71%) was indicated by a device with p(hPhDTP) [166].

Kranthiraja et al. used a π-conjugated polymer P-TT-TPD (poly[4,8-bis(2-(4-(2-ethylhexyloxy)phenyl)-5-thienyl)benzo[1,2-b:4,5-b′]dithiophene-alt-1,3-bis(6-octylthieno[3,2-b]thiophen-2-yl)-5-(2-hexyldecyl)-4H-thieno[3,4-c]pyrrole-4,6(5H)-dione]) for PSCs. The device of P-TT-TPD had a PCE of 16.82% and 17.09% in dopant-free and tris(pentafluorophenyl)borane-doped PSCs, respectively, due to P-TT-TPD good solution processability, well-suited energy levels, its high mobility, better passivation and high-dipole moment difference between the ground and excited states [167].

Kong et al. studied F-substituted benzodithiophene copolymers PBDT[2H]T, PBDT[2F]T, PBDT(T)[2F]T as dopant-free efficient HTMs in PSCs. The PSC of PBDT[2F]T was shown a PCE of 17.52%. The experiments revealed that PBDT[2F]T as an HTM could extract holes while concurrently passivating surface traps, making it a strong rival to the doped spiro-OMeTAD. Furthermore, the hydrophobic character of PBDT[2F]T was provided greater ambient stability [168].

You et al. reported polymeric HTMs PBDB-O, PBDB-T (alkoxy and thiophene as the side chain of BDT appropriately), and PBDB-Cz (carbazole as the conjugated side chains of BDT). PBDB-Cz had the highest HOMO energy level, hole mobility, passivation effect, and effective interface modification, all of which helped improve the V_oc_, J_sc_, and FF in the devices. The PSC with PBDB-Cz was the best-performing PCE at 21.11% [169].

Liu et al. presented a novel polymer P25NH (DPP-based donor−acceptor) for application as a HTM in PSCs. The P25NH exhibited high mobility, better aggregation than P3HT and stability at low concentrations, and a perovskite surface passivation effect. All of these benefits resulted in devices with a dopant-free low concentration of the P25NH with a comparatively high PCE (17.3%) [170]. Furthermore, Liu et al. synthesized a new P5NH compound analogous to the previous polymer P25NH. The polymer P5NH was demonstrated to have higher mobility (5.13 × 10^−2^ cm^−2^ V^−1^ s^−1^) than the reported P25NH (2.10 × 10^−2^ cm^−2^ V^−1^ s^−1^). The fabricated PSC with the P5NH was achieved at an efficiency of 18.1% [170,171].

### 4.3. Polymers as Additives of Electron Transport Layers (ETL) and Electron-Transporting Materials (ETM)

The ETL collects electrons from the perovskite layer/s and transports them into the external circuit in perovskite solar cells. As a result, an ideal ETL material should have high electron mobility and a perovskite-like energy level.

Xiong et al. used a P3HT (poly(3-hexylthiophene)) as an additive to the electron transport layer of PCBM ([6,6]-phenyl-C61-butyric acid methyl ester). The addition of P3HT to PCBM could enhance the surface morphology of ETL as well as the moisture and water resistance of the ETL. The findings suggested that a small amount of P3HT did not result in a decreased power conversion efficiency (PCE) and could increase the PCBM aggregation, resulting in an improved ETL moisture and water resistance [173].

Jiang et al. applied a doping PCBM with F8BT (poly(9,9-dioctylfluorene-co-benzothiadiazole) as the electron transport layer. Doping with F8BT resulted in the creation of a smooth and uniform ETL, which was beneficial for electron-hole pair separation and hence increased the PSC performance. The power conversion efficiency of 15% of the PSC with 5 wt.% F8BT in PCBM was achieved (Figure 6) [174].

Furthermore, You et al. introduced a biological polymer HP (heparin potassium) for stabilizing the ETL (SnO_2_) dispersion and depositing arrangement of ETL. This method was discovered to enhance the interface contact between the ETL and the perovskite layer by generating vertically aligned crystal growth of mixed-cation perovskites. The planar PSCs based on SnO_2_–HP had a PCE of over 23% (6 mg mL^−1^) on rigid substrates and 19.47% on flexible substrates [175].

Liu et al. studied BCP (bathocuproine)/PMMA (poly(methyl methacrylate)) and BCP/PVP (polyvinylpyrrolidone) films as hole-blocking/electron-transporting interfacial layers. The storage stability of PSCs with BCP/PMMA was greatly improved over the PSCs with BCP, but the photovoltaic performance was marginally reduced when PMMA was added. The increased hydrophobicity and moisture resistance of the resultant BCP/PMMA layer ensured better storage stability. The PVP enhanced electron transport over the BCP-based interfacial layer to the cathode, resulting in greater current densities and power efficiency in the devices (Figure 7) [176].

In addition, Said et al. investigated the impact of the sp^2^-N substitution position in the main chains of the polymeric compounds on the photovoltaic properties of devices. They employed pBTT, pBTTz, and pSNT (naphthalenediimide-based n-type polymers) as ETLs in PSCs. Adding sp^2^-N atoms to the donor thiophene units of pBTT resulted in pBTTz, which had somewhat lower electron mobility but greatly enhanced the PCE of PSCs. However, the PSC performance of pSNT with two extra sp^2^-N atoms and very high electron mobility was significantly worse. Furthermore, the electron-rich sulfur atoms had a significant influence on the passivating of the under-coordinated Pb-atoms, as evidenced by the current density–voltage (J–V) hysteresis curves of the devices with pBTTz [177].

Tian et al. reported n-type conjugated polymers with fluoro- and amino-side chains (PN, PN-F25%, and PN-F50%) as ETM in a perovskite device of structure ITO/NiO_x_/CH_3_NH_3_PbI_3−x_Cl_x_/PN or PN-F25% or PN-F50%/Ag. It was discovered that the amino side chains could provide good interface modification capabilities, while the fluoro side chains could supply hydrophobic qualities to these polymers. As a result, the bifunctional conjugated polymers successfully improved the performance of tested solar cells (17.5%), which was higher than the performance of devices with PC61BM (14.0%). Furthermore, the bifunctional ETMs were improved PSCs stability significantly [178].

Moreover, Elnaggar et al. tested a pyrrolo[3,4-c]pyrrole-1,4-dione-based n-type copolymer (P1, this polymer with the fullerene derivative [60]PCBM) as an electron transport material for PSCs. A conjugated polymer P1 and its composites with [60] PCBM provided reasonable efficiencies of 12–14%, respectively. Importantly, the use of the P1-PCBM [60] composite’s ETL resulted in a significant increase in the operational stability of PSCs [179].

Yan et al. synthesized semiconducting copolymers NDI-Se, NDI-BiSe, and NDI-TriSe based naphthalene-diimide. The addition of a biselenophene or triselenophene unit to a polymer increased the polymer’s planarity and delocalization, as well as conductivity. The perovskite solar cells of the ITO/NiO_x_/perovskite/NDI-selenophene/Ag structure were prepared. The power conversion efficiency of 9.51% (NDI-Se), 7.66% (NDI-BiSe), and 14.00% (NDI-TriSe) were obtained [180].

These studies contributed to the development of new polymeric ETLs and additives to the ETL by providing useful design recommendations.

### 4.4. Polymeric Interlayer/s

Interface engineering has been shown to be an effective method for reducing defect density in organic–inorganic hybrid PSCs and is commonly utilized to improve their performance [181,182,183,184,185,186,187,188].

Ding et al. exhibited a PVAc (polyvinyl acetate) as an agent modifying the surface of perovskite (CsPbBr_3_) film. The combination of O atoms in the carbonyl group (C=O) of PVAc with the positively charged under-coordinated Pb^2+^ and Cs^+^ defect ions could contribute to the reduction of the CsPbBr_3_ surface defects and improve the energy-level alignment between the carbon electrode (work function) and the valance band (VB) of perovskite. This results in reduced carrier recombination and energy loss at the perovskite/carbon contact, which effectively increased the V_oc_ and PCE [181].

Zhao et al. applied a DPP-DTT (poly(N-alkyldiketopyrrolo-pyrrole dithienylthieno[3,2-b]thiophene)) multifunctional passivation layer to receive the stable and highly efficient devices. By coordinate bonding between the atoms containing lone-pair electrons (sulphur, oxygen, and nitrogen) in DPP-DTT and the under the coordinate Pb atoms in perovskite, DPP-DTT acted as an efficient passivation layer to decrease defects on the perovskite surface. The DPP-DTT could function as a hole-selective layer because of its high hole mobility (~10 cm^2^ V^−1^ s^−1^) and acceptable valence band (VB) (−5.31 eV) between the perovskite (−5.67 eV) and spiro-OMeTAD (−5.22 eV) to efficiently increase hole extraction and transport. DPP-DTT as an ultra-hydrophobic agent, improved the perovskite stability [182].

Sharma et al. developed an n-type conjugated polymer with a naphthalene diimide core and a vinylene linker and oligo (ethylene glycol) (P2G) as a stable cathode interface layer (CIL). P2G was shown to be an effective CIL for lowering interfacial energy barriers in hybrid perovskite solar cells, with a PCE of 17.6% for MAPbI_3_-based p-i-n planar devices vs. 15% for reference devices. Because of the effectiveness of P2G CIL, there appeared to be a potential technique for creating alcohol/water-soluble polymer interlayers with desirable electrical and electronic characteristics [183].

Zhou et al. reported a photoinitiation-crosslinked zwitterionic polymer (Dex-CB-MA) as an interfacial layer played an important role in the perovskite device performance. Dex-CB-MA (dextran with carboxybetaine modified by methacrylate) was used as the interfacial layer between the PEDOT:PSS (as hole extraction layer, HEL) and the perovskite layer and to improve the morphology of the perovskite film. The Dex-CB-MA was discovered to generate an effective charge-transfer process in perovskite solar cells. As a consequence, when compared to PSCs based on the PEDOT:PSS HEL, the PEDOT:PSS/Dex-CB-MA HEL perovskite solar cells using the PEDOT:PSS/Dex-CB-MA HEL showed a 30% increase in power conversion efficiency [184].

Zhao et al. demonstrated a polymer-based difluorobenzothiadiazole (PffBT4T-C9C13) as the interfacial material for planar PSCs. The PffBT4T-C9C13 was deposited between the HTL (spiro-OMeTAD) and the perovskite absorber by utilizing a well-refined deposition technique. When the prepared polymer was decorated, a uniform perovskite layer with a large grain was formed. The PffBT4T-C9C13 has passivated the surface defects of perovskite film and also protected the film from water corrosion. At the interface, the charge collection was adequately suppressed, which helped with charge extraction and transport. As a consequence, the ITO/TiO_2_/MA_1−x_FA_x_PbI_3_/PffBT4T-C9C13/spiro-MeOTAD/Ag structure of the device had the best power conversion efficiency of 19.37% (0.50 mg mL^−1^ of PffBT4T-C9C13) [185].

Liu et al. reported a conjugated polyelectrolyte PTFTS (poly[N-(4-sulfonatophenyl)-4,4′-diphenylamine-alt-N-(p-trifluoromethyl)phenyl-4,4′-diphenylamine] sodium salt) as the interlayer between GO (graphene oxide) and the perovskite. The polymers’ interlayers were allowed for the identification of the recombination channels at the front-contact interface in the inverted-type planar PSCs. The sulfonate-charged PTFTS had the unanticipated benefit of causing strong contact between PTFTS as the binding force and GO, allowing for the development of a uniform interfacial layer onto GO using a simple wet-chemical method. As a result, the best PCE for the device with PTFTS was 18.39% [186].

Kim et al. utilized a PDMS (polydimethylsiloxane) interlayer between CuSCN (inorganic HTL) and the absorber to stabilize the perovskite deposition and working device. The PDMS successfully blocked a disintegration of perovskite at the surface throughout the upper layer’s deposition. In addition, it was noticed that the polymer could form chemical bonds with CuSCN and perovskite as the cross-linking interlayer. This novel cross-linking layer alleviated the interfacial traps/defects in the solar cells and amended the hole-extraction property at the interface. The PDMS as a cross-linking interlayer was enabled to receive a highly efficient PSC with MAPbI_3_/PDMS/CuSCN with a PCE of 19.04% [187].

Wang et al. synthesized a naphthalene imide dimer (2FBT2NDI) and applied it as interface material for inverted PSCs. With the introduction conjugated skeleton benzothiadiazole-dithiophene unit, two fluorine atoms could enhance intermolecular interactions and regulate the energy levels. The exploitation 2FBT2NDI as a polymeric interlayer suppressed the recombination of a charge trapped at the perovskite/ETL interface and improved the electron extraction and the efficiency of the device. The perovskite device with 2FBT2NDI exhibited the best PCE of 20.1% [188]. These works give precepts for implementing interface control and modification utilizing polymers.

The above-presented results prove that polymers can be utilized in PSCs to aid nucleation, control perovskite film crystallization, and improve device stability by developing different interactions with the perovskite films. The application of the DI monomer by adding it to the PbI_2_ precursor (polymerization-assisted grain growth (PAGG) technique) allowed for the appropriate interaction of the polymer with PbI_2_, thanks to which the PCE was obtained at the level of 23%, and the rate of perovskite degradation was slowed down [148]. Moreover, the utilization of the naphthalene imide dimer (2FBT2NDI) as interface material for reverse PSCs allowed for the inhibiting recombination of the charge trapped at the perovskite/ETL interface, improving the electron extraction as well as the device efficiency (20.1%) [188]. Additionally, polymers can also be shown high hole mobility, which makes their use possible as hole-transporting materials. The polymeric HTMs PBDB-Cz (carbazole as the conjugated side chains of BDT), was indicated a PCE of 21.11% [169]. Furthermore, biological polymer HP (heparin potassium) was introduced for stabilizing the electron transport layers’ dispersion and depositing the arrangement of the ETL (the PCE of over 23%) [175]. Consequently, it is imperative to design the novel polymers used in perovskite solar cells to improve the stability and performance of the PSCs.

## 5. Summary and Conclusions

Photovoltaics is a strongly and rapidly growing branch of renewable energy sources. Many new materials are being used in solar cells. Polymeric materials are also widely used in devices that convert solar radiation into electricity. As presented in this review paper, polymeric compounds are widely used in many fields in photovoltaic cells due to their numerous advantages, which undoubtedly include the possibility of modifying their chemical structure and thus adjusting their physical and chemical properties to the given needs. Polymeric materials are also widely used in devices that convert solar radiation into electricity. As presented in this review paper, polymeric compounds are widely used in many fields in photovoltaic cells due to their numerous advantages, which undoubtedly include the possibility of modifying their chemical structure and thus adjusting their physical and chemical properties to the given needs. Given current trends and recent literature, more and more new polymeric materials are finding applications in photovoltaic cells, as seen, for example, in their use as dyes in DSSCs or HTMs in PSCs.

In summary, polymeric materials are increasingly used in a wide range of research and technological solutions and will certainly become more widely and extensively used in solar cells as well. As noted, polymers are used as the flexible transparent substrates for all types of photovoltaic devices discussed, as materials that impart gel character to electrolytes in DSSCs, counter-electrodes, materials responsible for the pore formation in inorganic oxides used in DSSCs and PSCs. They are widely used also in BHJ, mainly as donor materials, but numerous studies report that the substitution of acceptor fullerenes by polymers can also be found. It is also worth remembering to use polymers as intermediate or buffer layers, supporting the transport or separation of the generated charges to the appropriate electrodes. As previously mentioned, organic compounds and among them, polymers, are and will be widely used in new technologies for obtaining electric currents due to their relatively low costs of preparation, easy modification of the chemical structure, and thus easily obtain the required properties, as well as the possibility of manufacturing suitable layers from them.

As mentioned in the summaries of the individual sections, it is not easy or even entirely possible to compare and determine the best polymer on the basis of the collected results due to the differences in structures and methods of preparation of individual devices. In the case of the DSSC cells, apart from the use of the polymer as a dye or counter-electrode, it is very difficult to determine the direct effect on the recorded device parameters. Even in the case of a dye or counter-electrode, the preparation methods and conditions play a huge role, with very often different results in different publications. Therefore, it can only be stated that the following polymers are used PEN, PET as flexible substrates; PS, PVP, P123 used to obtain pores in the oxide material, PEO, PAN, PEG as materials to give a gel structure to the electrolyte and PANI, PPy, PTh, PEDOT, PEDOT:PSS as counter-electrodes. In the case of BHJ solar cells, high efficiencies are registered mainly for donors containing thiophene rings in the polymer, such as PTB-7, PTB-7-Th, D18, P106, among others. As for acceptors, polymeric compounds are used much less frequently, while high efficiencies are registered for polymers containing naphthalene moieties. In perovskite solar cells, polymers are used as additives to facilitate the nucleation and crystallization processes in the perovskite layer(s). By adding a DI monomer to the PbI_2_ precursor, a PCE was obtained of 23.0%. Polymers can also be used as electrons (biological polymer HP, the PCE of over 23%), hole-transporting materials (PBDB-Cz, PCE = 21.11%), and interface layer(s) (2FBT2NDI, PCE = 20.1%).

## Figures and Tables

**Figure 1 polymers-14-01946-f001:**
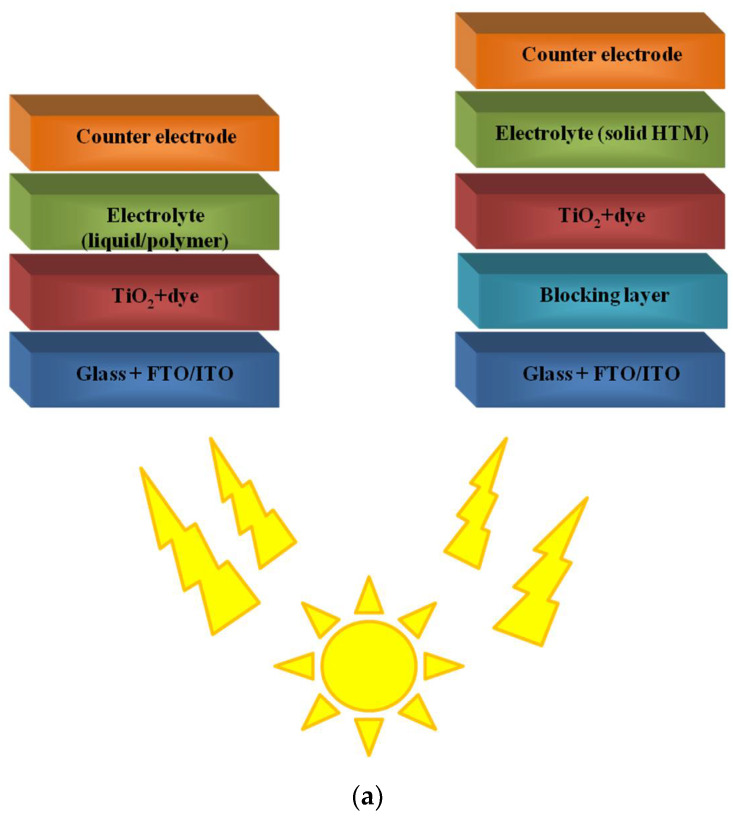
The structures of DSSCs with liquid and solid electrolyte (**a**) and polymer structures often used in this type of solar cell (**b**).

**Figure 2 polymers-14-01946-f002:**
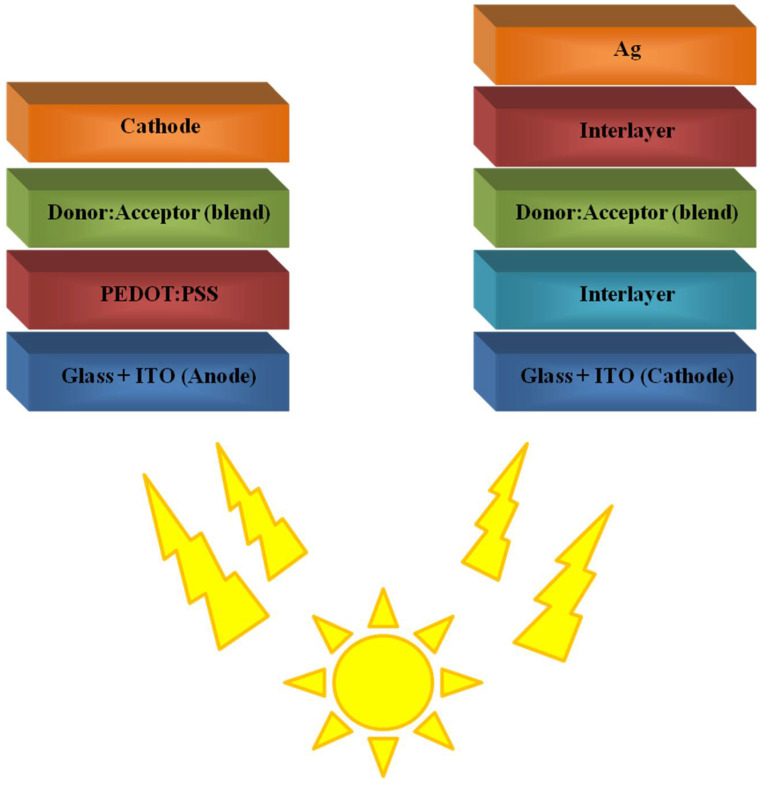
Schematic structure of a BHJ OSC with the conventional and inverted systems.

**Figure 3 polymers-14-01946-f003:**
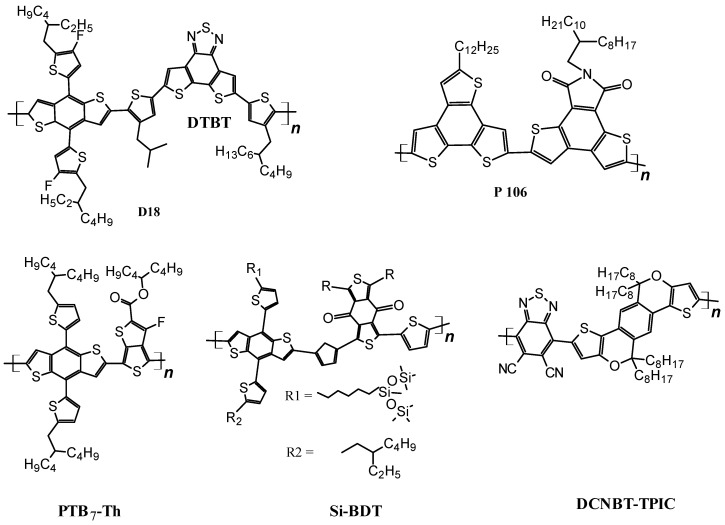
Chemical structures of wide bandgap polymers used in BHJ solar cells.

**Figure 4 polymers-14-01946-f004:**
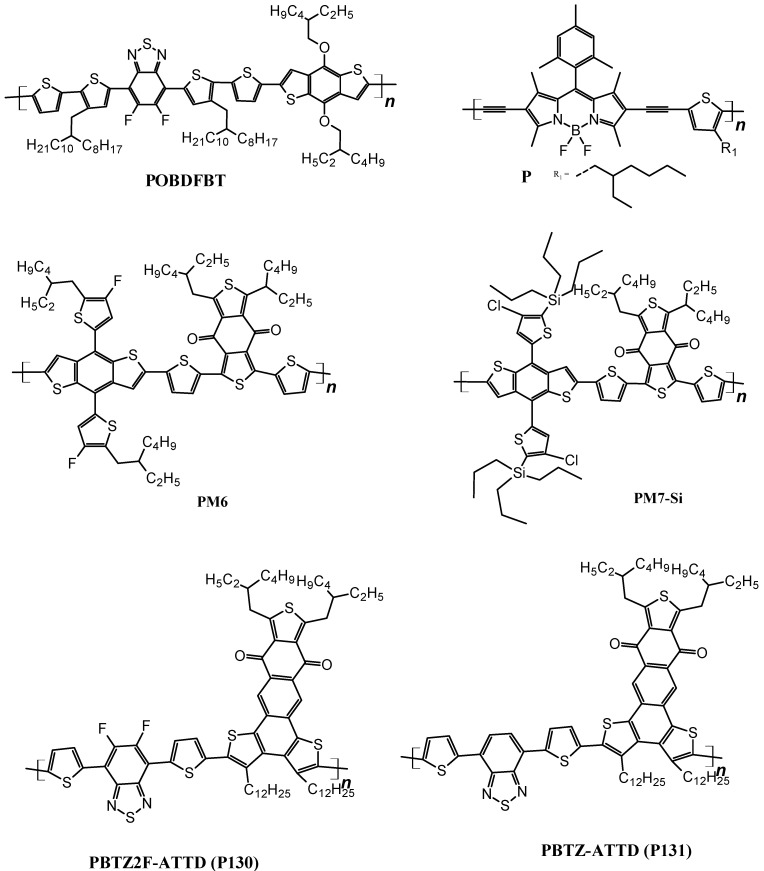
Chemical structures of medium bandgap polymers used in BHJ solar cells.

**Figure 5 polymers-14-01946-f005:**
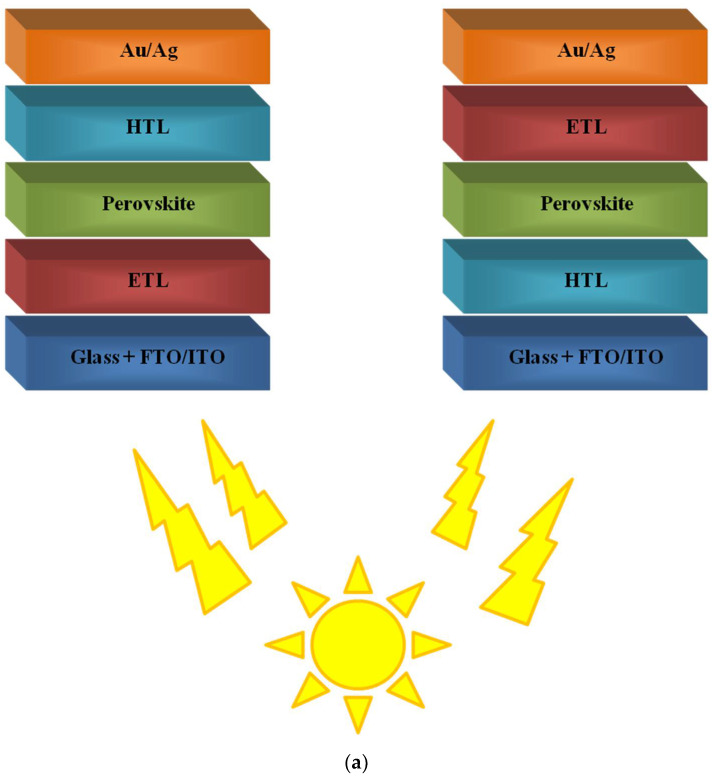
The conventional and inverted architecture of perovskite solar cells (**a**) and the chemical structures of polymers used in PSCs (**b**).

**Figure 6 polymers-14-01946-f006:**
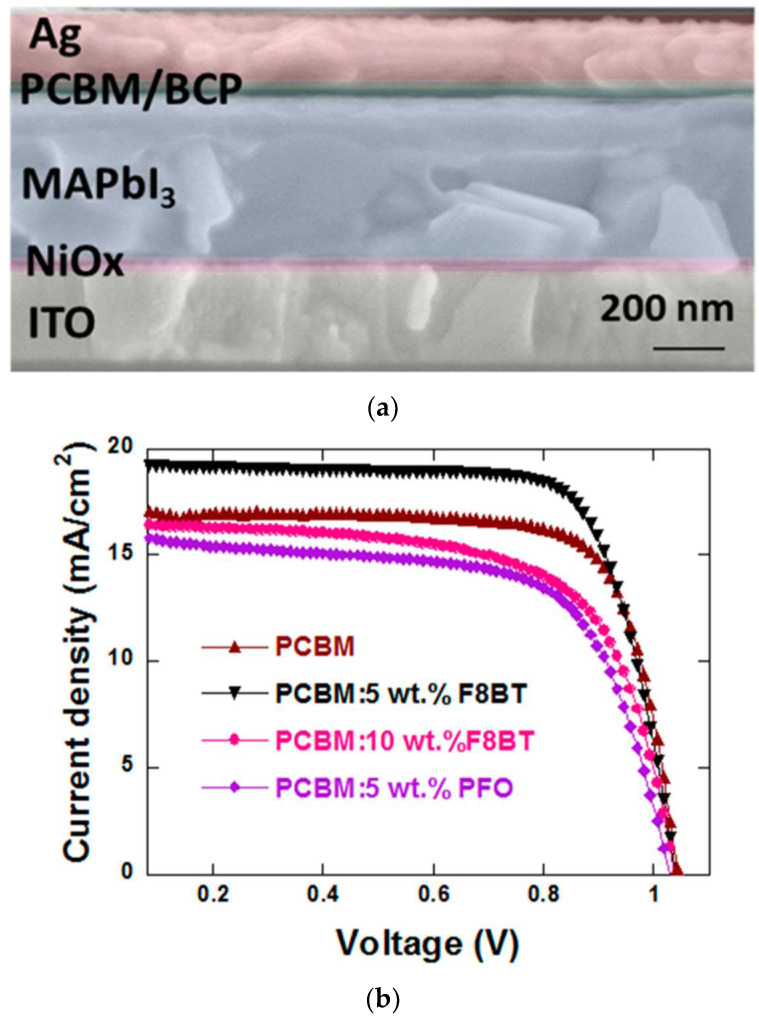
The sectional SEM image of PSC (**a**) and photocurrent density–voltage curves of the devices with F8BT in PCBM (**b**) [174].

**Figure 7 polymers-14-01946-f007:**
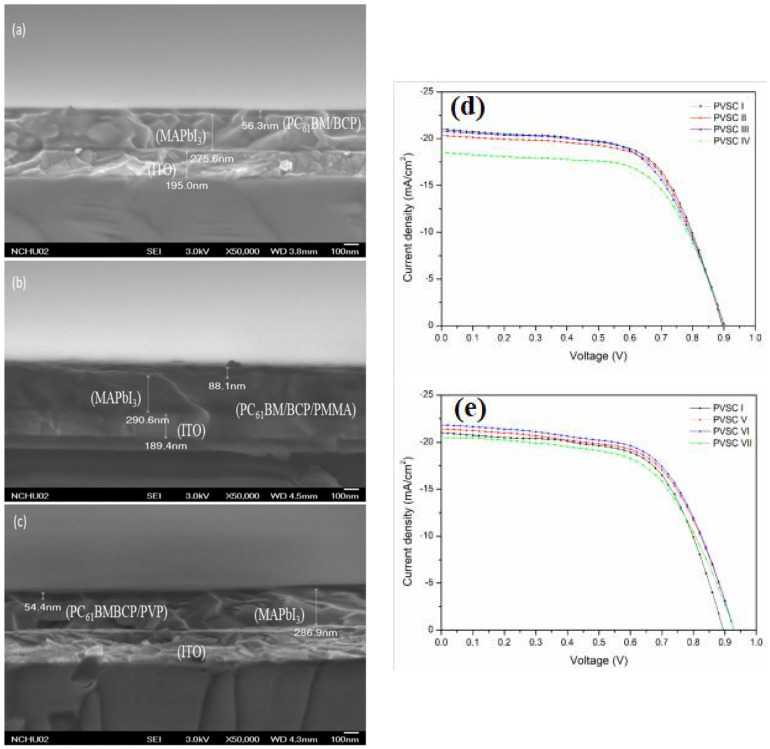
The cross-sectional SEM images of ITO/PEDOT:PSS/MAPbI_3_/PC_61_BM with BCP (**a**), BCP/PMMA (**b**), BCP/PVP (**c**) and the current density–potential plots of PSCs with BCP (PVSC I) (**d**,**e**), BCP/PMMA (PVSC II–IV) (**d**), BCP/PVP (PVSC V–VII)) (**e**) [176].

**Table 1 polymers-14-01946-t001:** Photovoltaic parameters of flexible dye-sensitized solar cells, containing dye N719.

Substrate	V_oc_ (mV)	J_sc_ (mA/cm^2^)	FF (-)	PCE (%)	Ref.
ITO/PEN	660	8.97	0.45	2.65	[26]
ITO/PEN + G LSL	680	10.62	0.48	3.47	
ITO/PEN + T LSL	690	14.65	0.43	4.33	
ITO/PEN + TG LSL	680	14.32	0.53	5.18	
ITO/PEN (100 mW/cm^2^)	400	8.70	0.46	2.60	[68]
ITO/PEN (18 mW/cm^2^)	380	1.8	0.53	3.30	
ITO/PET (Pt CE)	685	5.53	0.78	2.95	[69]
ITO/PET (PEDOT:PSS CE)	695	6.09	0.52	2.18	

T—titanium dioxide, G—grapheme, TG—titanium dioxide-graphene, LSL—light-scattering layer.

**Table 2 polymers-14-01946-t002:** Photovoltaic parameters of quasi-solid-state DSSC, based on the commercial Ru-dye.

Gel Electrolyte Structure	Solvent	V_oc_ (mV)	J_sc_ (mA/cm^2^)	FF (-)	PCE (%)	Ref.
PEO + PGEDME/I^−^/I_3_^−^ (0.1 M)	EtOH	730	12.91	0.66	6.34	[34]
	ACN	752	13.45	0.67	6.87	
	ACN/VN	785	11.56	0.76	6.88	
	ACN/3-MPN	785	12.05	0.76	7.16	
PEO + PGEDME/I^−^/I_3_^−^ (0.2 M)	ACN/VN	785	12.49	0.75	7.39	
PEO + PGEDME/I^−^/I_3_^−^ (0.4 M)		784	11.94	0.75	7.03	
PEO + PGEDME/I^−^/I_3_^−^ (0.2 M) + GuSCN (0 M)		778	11.94	0.75	7.00	
PEO + PGEDME/I^−^/I_3_^−^ (0.2 M) + GuSCN (0.05 M)		787	12.48	0.75	7.35	
PEO + PGEDME/I^−^/I_3_^−^ (0.2 M) + GuSCN (0.1 M)		793	12.56	0.77	7.66	
PEO + PGEDME/I^−^/I_3_^−^ (0.2 M) + GuSCN (0.2 M)		758	12.85	0.73	7.07	
PAN (I^−^/I_3_^−^)	DMF	790	6.85	0.67	4.19	[35]
C11-AZO-C11/PAN (I^−^/I_3_^−^)		780	11.96	0.75	6.28	
3 wt.% PAN-co-PBA (I^−^/I_3_^−^)	ACN	593	9.86	0.64	3.77	[36]
5 wt.% PAN-co-PBA (I^−^/I_3_^−^)		587	11.60	0.61	4.13	
7 wt.% PAN-co-PBA (I^−^/I_3_^−^)		646	13.16	0.62	5.23	
9 wt.% PAN-co-PBA (I^−^/I_3_^−^)		618	10.41	0.65	4.35	
**10 wt.% PVdF-HFP (I^−^/I_3_^−^)**	ACN	6920	10.34	0.66	4.74	[77]
9 wt.% PVdF-HFP (I^−^/I_3_^−^)		690	13.75	0.63	6.02	
8 wt.% PVdF-HFP (I^−^/I_3_^−^)		670	12.04	0.62	5.03	
7 wt.% PVdF-HFP (I^−^/I_3_^−^)		660	10.04	0.58	3.97	
**pCMA-PGE (I^−^/I_3_^−^)**	PC:ACN	545	10.30	0.34	2.20	[78]
PVP-PGE (I^−^/I_3_^−^)		640	6.67	0.60	3.00	
PVDF (I^−^/I_3_^−^)	ACN	730	17.79	0.64	8.36	[79]
PDA@PVDF (I^−^/I_3_^−^)		720	17.95	0.64	8.26	
esPME (I^−^/I_3_^−^)	ACN	710	13.10	0.69	6.42	[80]
esCPME (2 wt.% PPy) (I^−^/I_3_^−^)		720	13.90	0.70	7.02	

EtOH—ethanol, ACN—acetonitrile, VN—valeronitrile, 3-MPN—3-methoxypropionitrile, GuSCN—guanidine thiocyanate, DMF—N,N-Dimethylformamide, PC—propylene carbonate.

**Table 3 polymers-14-01946-t003:** Photovoltaic performance parameters of polypyrrole-based DSSCs counter-electrodes and their nanocomposites.

Counter-Electrodes	Fabrication Methods of PPy-Based Counter-Electrodes	V_oc_ (mV)	J_sc_ (mA cm^−2^)	FF (-)	PCE (%)	Ref.
Polypyrrole	Doctor Blade Technique	749	15.75	0.69	8.13	[37]
Poylpyrrole	Electropolymerization	727	10.20	0.42	3.12	[100]
Polypyrrole	Electropolymerization	630	12.00	0.51	3.86	[101]
PPy-POM	Electropolymerization	765	11.68	0.56	5.04	[38]
PPy-SrTiO_3_	Doctor Blade Technique	671	10.45	0.36	2.52	[103]
PPy-MoS	In-situ Polymerization	708	18.90	0.62	8.28	[105]

**Table 4 polymers-14-01946-t004:** Photovoltaic performance parameters of polyaniline and its nanocomposites based on DSSCs counter-electrodes.

Counter-Electrode	Fabrication Methods of PANI-Based Electrodes	V_oc_ (mV)	J_sc_ (mA cm^−2^)	FF (-)	PCE (%)	Ref.
PANI	Doctor Blade Technique	645	20.8	0.41	4.20	[39]
PANI	Screen Printing Technique	630	5.10	0.48	1.71	[106]
PANI	Cyclic Voltametric-Electrochemical Method	740	15.34	0.64	7.27	[107]
Pristine PANI	Doctor Blade Technique	480	4.71	0.45	1.14	[40]
H_2_SO_4_-doped PANI	Doctor Blade Technique	530	7.86	0.43	1.78	
ALS-doped PANI	Doctor Blade Technique	603	10.84	0.43	2.79	
ALS-H_2_SO_4_-doped PANI	Doctor Blade Technique	603	15.13	0.53	4.54	
WO_3_-PANI	Cyclic voltammetry Technique	685	18.00	0.55	6.78	[108]

**Table 5 polymers-14-01946-t005:** Photovoltaic parameters of devices containing wide-bandgap polymers.

Structure of Solar Cell	V_oc_ (mV)	J_sc_ (mA/cm^2^)	FF (-)	PCE (%)	Ref.
ITO/PEDOT:PSS/W1:Y6 (1:1)/PDIN/Ag	890	25.36	0.68	15.39 (14.95) *^a^*	[121]
ITO/PEDOT:PSS/W1:Y6 (1:1.2)/PDIN/Ag	890	25.92	0.69	15.95 (15.69)	
ITO/PEDOT:PSS/W1:Y6 (1:1.4)/PDIN/Ag	890	25.06	0.71	15.87 (15.64)	
ITO/PEDOT:PSS/W1:Y6 (1:1.6)/PDIN/Ag	880	24.59	0.71	15.65 (15.35)	
ITO/PEDOT:PSS/PBDT- TTZ:N2200/PFN-Br/Ag	870	14.4	0.67	8.40	[122]
ITO/PEDOT:PSS/PBDT-TT:N2200/PFN-Br/Ag	750	2.0	0.46	0.70	
ITO/PEDOT:PSS/PBDT-TTz:PC61BM/PFN-Br/Ag	890	10.3	0.73	6.70	
ITO.PEDOT:PSS/PBTz:IT-4F/PFN-Br/Al	840	17.68	0.59	8.76	[123]
ITO/PEDOT:PSS/PTzTz:IT-4F/PFN-Br/Al	820	18.81	0.69	10.63	
ITO/PEDOT:PSS/D18:Y6 (1:0.8)/PDIN/Ag	861	27.16	0.72	16.98 (16.76) *^a^*	[15]
ITO/PEDOT:PSS/D18:Y6 (1:1.2)/PDIN/Ag	863	27.05	0.75	17.51 (17.34)	
ITO/PEDOT:PSS/D18:Y6 (1:1.6)/PDIN/Ag	865	27.31	0.75	17.84 (17.67)	
ITO/PEDOT:PSS/D18:Y6 (1:2)/PDIN/Ag	870	26.20	0.75	17.16 (16.87)	
ITO/PEDOT:PSS/D18:Y6/PDIN/Ag (170 nm) *^b^*	864	25.89	0.73	16.38 (16.34) *^a^*	
ITO/PEDOT:PSS/D18:Y6/PDIN/Ag (130 nm)	864	26.39	0.74	16.92 (16.77)	
ITO/PEDOT:PSS/D18:Y6/PDIN/Ag (112 nm)	866	27.16	0.75	17.65 (17.38)	
ITO/PEDOT:PSS/D18:Y6/PDIN/Ag (103 nm)	865	27.31	0.75	17.84 (17.67)	
ITO/PEDOT:PSS/D18:Y6/PDIN/Ag (91 nm)	869	26.75	0.76	17.73 (17.64)	
ITO/PEDOT:PSS/D18:Y6:PC61BM (1:1.6:0)/PDIN/Ag	862	26.09	0.76	17.23 (17.01) *^b^*	[124]
ITO/PEDOT:PSS/D18:Y6:PC61BM (1:1.6:0.1)/PDIN/Ag	865	26.33	0.76	17.42 (17.16)	
ITO/PEDOT:PSS/D18:Y6:PC61BM (1:1.6:0.2)/PDIN/Ag	870	26.48	0.77	17.89 (17.57)	
ITO/PEDOT:PSS/D18:Y6:PC61BM (1:1.6:0.4)/PDIN/Ag	874	25.70	0.75	16.94 (16.78)	
ITO/PEDOT:PSS/D18:Y6:PC61BM (1:1.6:0.6)/PDIN/Ag	882	25.64	0.71	16.05 (15.82)	
ITO/PEDOT:PSS/D18:Y6:PC61BM (1:1.6:0.2 *^c^*)/PDIN/Ag	865	25.90	0.76	17.09 (16.92)	
ITO/PEDOT:PSS/D18:Y6:PC61BM (1:1.6:0 *^d^*)/PDIN/Ag	865	27.31	0.75	17.84 (17.67)	
ITO/PEDOT:PSS/D18:Y6:PC61BM (1:1.6:0 *^e^*)/PDIN/Ag	859	27.70	0.76	18.22 (18.01)	
ITO/PEDOT:PSS/D18:Y6:PC61BM/PDIN/Ag (90 nm) *^b^*	870	25.94	0.75	17.10 (16.97) *^b^*	
ITO/PEDOT:PSS/D18:Y6:PC61BM/PDIN/Ag (110 nm)	870	26.48	0.77	17.89 (17.57)	
ITO/PEDOT:PSS/D18:Y6:PC61BM/PDIN/Ag (130 nm)	864	26.44	0.75	17.12 (16.72)	
ITO/PEDOT:PSS/PTPD:Y6/PFN-Br/Ag	660	19.5	0.46	5.90	[125]
ITO/PEDOT:PSS/PBiTPD:Y6/PFN-Br/Ag	830	25.6	0.66	14.20	
ITO/PEDOT:PSS/P106:Y18-DMO/PFN/Al	870	22.78 (22.62) *^f^*	0.71	14.07 (13.91) *^b^*	[126]
ITO/PEDOT:PSS/P106:DBTBT-IC/PFN/Al	960	18.56 (18.41)	0.66	11.76 (11.59)	
ITO/PEDOT:PSS/P106:DBTBT-IC:Y18-DMO/PFN/Al	910	24.82 (24.66)	0.73	16.49 (16.32)	
ITO/ZnO/[PTB7-Th(1):Si-BDT(0):DCNBT-TPIC(0.6)/MoO_3_/Ag	850	18.00 (18.07) *^f^*	0.64	10.11	[127]
ITO/ZnO/[PTB7-Th(0.8):Si-BDT(0.2):DCNBT-TPIC(0.6)/MoO_3_/Ag	850	19.32 (19.74) *^f^*	0.65	11.20	
ITO/ZnO/[PTB7-Th(0.6):Si-BDT(0.4):DCNBT-TPIC(0.6)/MoO_3_/Ag	860	22.32 (22.06) *^f^*	0.68	13.45	
ITO/ZnO/[PTB7-Th(0.4):Si-BDT(0.6):DCNBT-TPIC(0.6)/MoO_3_/Ag	820	19.21 (19.23) *^f^*	0.66	10.88	
ITO/ZnO/[PTB7-Th(0.2):Si-BDT(0.8):DCNBT-TPIC(0.6)/MoO_3_/Ag	820	16.00 (16.01) *^f^*	0.54	7.53	
ITO/ZnO/[PTB7-Th(0):Si-BDT(1):DCNBT-TPIC(0.6)/MoO_3_/Ag	820	17.58 (17.59) *^f^*	0.65	9.92	

*^a^* Data in parentheses are averages for 10 cells; *^b^* Data in parentheses are the thickness of the active layer; *^c^* The active layer underwent CF solvent vapor annealing (SVA) for 5 min; *^d^* No CN additive; *^e^* No CN additive; the active layer underwent CF SVA for 5 min. Data from literature, *^f^* Estimated from the integration of EQE spectra.

**Table 6 polymers-14-01946-t006:** Photovoltaic parameters of devices containing medium-bandgap polymers.

Structure of Solar Cell	V_oc_ (mV)	J_sc_ (mA/cm^2^)	FF (-)	PCE (%)	Ref.
ITO/PEDOT:PSS/POBDFBT(1):ITIC(1): PCBM(0)/PFN/Al	820	16.59	0.46	6.16	[130]
ITO/PEDOT:PSS/POBDFBT(1):ITIC(1): PCBM(0.5)/PFN/Al	780	12.7	0.64	6.26	
ITO/PEDOT:PSS/POBDFBT(1):ITIC(0.75): PCBM(0.75)/PFN/Al	760	13.8	0.61	6.39	
ITO/PEDOT:PSS/POBDFB(1):ITIC(0.5): PCBM(1)/PFN/Al	720	17.65	0.62	7.91	
ITO/PEDOT:PSS/POBDFB(1):ITIC(0.25): PCBM(1.25)/PFN/Al	790	13.78	0.61	6.66	
ITO/PEDOT:PSS/POBDFB(1):ITIC(0): PCBM(1.5)/PFN/Al	710	13.67	0.64	6.23	
ITO/PEDOT:PSS/P:ITIC-m/PFN/Al	1040	16.86	0.69	12.10	[131]
ITO/PEDOT:PSS/P:Y6/PFN/Al	940	19.72	0.71	13.16	
ITO/PEDOT:PSS/P:ITIC-m:Y6/PFN/Al	990	20.65	0.74	15.13	
ITO/PEDOT:PSS/PM6:MF1(0):Y6/PDIN/Al	843	25.11	0.75	15.93	[132]
ITO/PEDOT:PSS/PM6:MF1(10):Y6/PDIN/Al	853	25.68	0.77	17.22	
ITO/PEDOT:PSS/PM6:MF1(50):Y6/PDIN/Al	867	23.53	0.71	14.40	
ITO/PEDOT:PSS/PM6:MF1(100):Y6/PDIN/Al	914	16.67	0.79	12.09	
ITO/PEDOT:PSS/PM6:Y6/PDINO/Al (150)	860	24.3	0.73	15.3(15.2 ± 0.1)	[133]
ITO/PEDOT:PSS/PM6:Y6/PDINO/Al (150)	830	25.3	0.75	15.7(15.6 ± 0.1)	
ITO/PEDOT:PSS/PM6:Y6/PDINO/Al (200)	830	25.8	0.67	14.3(14.2 ± 0.1)	
ITO/PEDOT:PSS/PM6:Y6/PDINO/Al (250)	820	27.1	0.63	14.1(13.9 ± 0.2)	
ITO/PEDOT:PSS/PM6:Y6/PDINO/Al (300)	820	26.5	0.62	13.6(13.3 ± 0.3)	
ITO/ZnO/PM6:Y6/MoO_3_/Ag (100)	820	25.2	0.76	15.7(15.5 ± 0.2)	
ITO/PEDOT:PSS/PM6:Y6/PDINO/Al	830	23.2	0.77	14.90	
ITO/PEDOT:PSS/PM6(1):Y6 (1.2): PC_71_BM(0)/PDINO/Al	8450	24.89	0.74	15.75 (15.70)	[134]
ITO/PEDOT:PSS/PM6(1):Y6 (1.1): PC_71_BM(0.1)/PDINO/Al	850	25.36	0.76	16.30 (16.26)	
ITO/PEDOT:PSS/PM6(1):Y6 (1.05): PC_71_BM(0.15)/PDINO/Al	850	25.8	0.75	16.38 (16.32)	
ITO/PEDOT:PSS/PM6(1):Y6 (1.0): PC_71_BM(0.2)/PDINO/Al	850	25.7	0.76	16.67 (16.61)	
ITO/PEDOT:PSS/PM6(1):Y6 (0.9): PC_71_BM(0.3)/PDINO/Al	853	25.05	0.75	16.05 (16.0)	
ITO/PEDOT:PSS/PM6(1):Y6 (0.7): PC_71_BM(0.5)/PDINO/Al	865	23.94	0.74	15.30 (15.23)	
ITO/PEDOT:PSS/PM6(1):Y6 (0.4): PC_71_BM(0.8)/PDINO/Al	876	19.24	0.49	8.39 (8.27)	
ITO/PEDOT:PSS/PM6(1):Y6 (0.1): PC_71_BM(1.2)/PDINO/Al	965	11.56	0.53	6.01 (5.94)	
ITO/PEDOT:PSS/PM6(1):PM7-Si(0):C9(1.2)/PFN-Br/Ag	841	26.36	0.76	17.0	[135]
ITO/PEDOT:PSS/PM6(0.9):PM7-Si(0.1):C9(1.2)/PFN-Br/Ag	864	26.35	0.77	17.7	
ITO/PEDOT:PSS/PM6(0):PM7-Si(1):C9(1.2)/PFN-Br/Ag	895	14.43	0.41	5.4	
ITO/PEDOT:PSS/P130:Y6/PFN/Al	890 (±5)	23.84 (±0.32)	0.72 (±0.05)	15.28 (±0.21)	[136]
ITO/PEDOT:PSS/P131:Y6/PFN/Al	780 (±3)	21.96 (0.22)	0.65 (±0.03)	11.13 (±0.18)	

**Table 7 polymers-14-01946-t007:** Photovoltaic parameters of solar cells contain narrow bandgap polymer donors.

Device Structure	Voc (mV)	J_sc_ (mA cm^−2^)	FF (-)	PCE (%)	Ref.
**ITO/PEDOT:PSS/PffBT-DPP(1)/[70] PCBM(3)/MeIC(1)/ZrAcAc/Al**	740	12.5	0.74	6.8	[137]
ITO/PEDOT:PSS/PffBT-DPP(1)/[70] PCBM(0)/MeIC(1)/ZrAcAc/Al	780	4.5	0.58	2.0	
ITO/PEDOT:PSS/PffBT-DPP(1)/[70] PCBM(2)/MeIC(1)/ZrAcAc/Al	760	16.1	0.73	9.0	
(ITO)/PEDOT:PSS/PTQ10:Y6/PFN-Br/Al	820 ± 1	23.9 ± 0.1	0.73	14.5 ± 0.1	[138]
ITO/PEDOT:PSS/P1(1):PC_71_BM(2)/LiF/Al (500 rpm) ^i^	770	5.76	0.43	1.92	[139]
ITO/PEDOT:PSS/P1(1):PC_71_BM(3)/LiF/Al (500 rpm) ^i^	770	7.32	0.39	2.21	
ITO/PEDOT:PSS/P1(1):PC_71_BM(4)/LiF/Al (500 rpm) ^i^	770	7.10	0.39	1.97	
ITO/PEDOT:PSS/P1(1):PC_71_BM(3)/LiF/Al (500 rpm) ^i^	580	3.07	0.30	0.55	
ITO/PEDOT:PSS/P1(1):PC_71_BM(3)/LiF/Al (500 rpm) ^i^	770	8.19	0.35	2.21	
ITO/PEDOT:PSS/P1(1):PC_71_BM(3)/LiF/Al (350 rpm)^i^	790	7.29	0.41	2.36	
ITO/PEDOT:PSS/P1(1):PC_71_BM(3)/LiF/Al (750 rpm) ^i^	790	6.94	0.35	1.92	
ITO/PEDOT:PSS/P2(1):PC_71_BM(2)/LiF/Al (500 rpm) ^i^	710	5.27	0.55	2.07	
ITO/PEDOT:PSS/P2(1):PC_71_BM(3)/LiF/Al (500 rpm) ^i^	700	5.30	0.37	1.38	
ITO/PEDOT:PSS/P3(1):PC_71_BM(1)/LiF/Al (500 rpm) ^i^	750	2.50	0.49	0.92	
ITO/PEDOT:PSS/P3(1):PC_71_BM(2)/LiF/Al (500 rpm) ^i^	750	3.95	0.46	1.38	
ITO/PEDOT:PSS/P3(1):PC_71_BM(3)/LiF/Al (500 rpm) ^i^	750	3.85	0.49	1.43	
ITO/PEDOT:PSS/P3(1):PC_71_BM(4)/LiF/Al (500 rpm) ^i^	760	5.14	0.42	1.65	
ITO/PEDOT:PSS/P3(1):PC_71_BM(4)/LiF/Al (500 rpm) ^i^	740	7.13	0.34	1.83	
ITO/PEDOT:PSS/P3(1):PC_71_BM(4)/LiF/Al (500 rpm)^i^	750	7.63	0.35	2.02	
ITO/PEDOT:PSS/P3(1):PC_71_BM(4)/LiF/Al (350 rpm) ^i^	770	7.59	0.41	2.45	
ITO/PEDOT:PSS/P3(1):PC_71_BM(4)/LiF/Al (750 rpm) ^i^	740	5.9	0.33	1.48	
ITO/PEDOT:PSS/PTT-EFQX:PCBM/PFN-Br/Ag	690	11.19	0.68	5.37	[140]
**ITO/PEDOT:PSS/PT-DFBT-T-EFQX:PCBM/PFN-Br/Ag**	870	5.62	0.54	2.69	
**ITO/PEDOT:PSS/P(T2BDY−TBDT)/PNDIT-F3N−Br/Ag**	780	12.07	0.47	4.40	[141]
ITO/PEDOT:PSS/P(TTzBDY−TBDT)/PNDIT-F3N−Br/Ag	800	7.71	0.40	2.49	
ITO/PEDOT:PSS/P(T2BDY−TBDT0.7−OBDT0.3)/PNDIT-F3N−**Br/Ag**	750	3.80	0.37	1.06	
ITO/PEDOT:PSS/P(TTzBDY−TBDT0.7−OBDT0.3)/PNDIT-F3N−**Br/Ag**	770	5.23	0.39	1.58	

^i^ Revolutions per minute.

**Table 8 polymers-14-01946-t008:** Polymers used as an acceptor material in bulk-heterojunction solar cells.

Structure of Solar Cell	V_oc_ (mV)	J_sc_ (mA/cm^2^)	FF (-)	PCE (%)	Ref.
ITO/PEDOT:PSS/PTzBISi:N2200/C60N/Ag CB- as print	930	2.76	0.43	1.01	[47]
**ITO/PEDOT:PSS/PTzBISi:N2200/C60N/Ag CB-TA**	890	3.98	0.48	1.57	
**ITO/PEDOT:PSS/PTzBISi:N2200/C60N/Ag CB-TA+SVA**	870	4.58	0.51	1.83	
**ITO/PEDOT:PSS/PTzBISi:N2200/C60N/Ag MTHF-as print**	890	15.41	0.70	9.01	
**ITO/PEDOT:PSS/PTzBISi:N2200/C60N/Ag MTHF-TA**	880	16.19	0.73	9.96	
**ITO/PEDOT:PSS/PTzBISi:N2200/C60N/Ag MTHF-TA+SVA**	880	17.62	0.76	11.25	
**ITO/ZnO/PTB7-Th:NDP-V/V_2_O_5_/Al**	740	17.07	0.67	8.59	[48]
ITO/ZnO/PTB7-Th:PDI-V/V_2_**O_5_/Al**	740	15.39	0.64	7.38	
**ITO/ZnO/PEI/BSS0:PBDB-T/MoO_3_/Ag**	820	15.74	0.57	7.38	[49]
**ITO/ZnO/PEI/BSS10:PBDB-T/MoO_3_/Ag**	860	18.55	0.64	10.10	
**ITO/ZnO/PEI/BSS20:PBDB-T/MoO_3_/Ag**	860	17.07	0.65	9.58	
**ITO/ZnO/PEI/BSS50:PBDB-T/MoO_3_/Ag**	850	17.50	0.65	9.69	
ITO/ZnO/PBDBT:PIID(CO) 2FT/MoO_3_/**Ag**	640	8.30	0.50	2.65	[142]
ITO/ZnO/**PBDBT:PIID(CO) BTIA/MoO_3_/Ag**	630	1.80	0.50	0.37	
ITO/PEDOT:PSS/PBDB-Tb-PYT/PDINN50/Ag (CF; area 5 mm^2^)	919	16.90	0.46	7.18	[143]
ITO/PEDOT:PSS/PBDB-Tb-PYT/PDINN50/Ag (CF, 4% CN; area 5 mm^2^)	916	19.60	0.63	11.32	
ITO/PEDOT:PSS/PBDB-Tb-PYT/PDINN50/Ag (CB, 4% CN; area 5 mm^2^)	908	19.31	0.60	10.53	
ITO/PEDOT:PSS/PBDB-Tb-PYT/PDINN50/Ag (ODCB, 4% CN; area 5 mm^2^)	917	18.67	0.59	10.08	
ITO/PEDOT:PSS/PBDB-Tb-PYT/PDINN50/Ag (THF, 4% CN; area 5 mm^2^)	914	19.25	0.63	11.13	
ITO/PEDOT:PSS/PBDB-Tb-PYT/PDINN50/Ag (Toluene, 4% CN; area 5 mm^2^)	912	19.38	0.62	11.07	
ITO/PEDOT:PSS/PBDB-Tb-PYT/PDINN50/Ag (CF, 4% CN; area 2.2 mm^2^)	867	19.71	0.63	10.80	
ITO/PEDOT:PSS/PBDB-T:PYT/PDINN50/Ag (CF; area 5 mm^2^)	883	22.70	0.72	14.57	

MTHF—2-methyltetrahydrofuran, TA—thermal annealing, SVA—solvent vapor annealing, CF—chloroform, CB—chlorobenzene, ODCB—o-dichlorobenzene, CN—1-chloronaphthalene.

**Table 9 polymers-14-01946-t009:** The collected photovoltaic parameters of PSCs based on the analysed HTMs.

Device Structure	Voc (mV)	J_sc_ (mA cm^−2^)	FF (-)	PCE (%)	Ref.
**FTO/TiO_2_/SnO_2_/[Cs_0_._05_FA_0_._8_MA_0_._15_PbI_2_._55_Br_0_._45_]/IDTB/Au**	1107	23.06	0.76	19.38	[160]
**ITO/P3CT/[(FA_0.17_MA_0.94_PbI_3.11_)_0.95_(PbCl_2_)_0.05_]/C_60_/ZrAcac/Ag**	1120	22.88	0.82	21.09	[161]
**FTO/TiO_2_/[0.001 M FAI, 0.001 M PbI_2_, 0.0002 M MABr, 0.0002 M PbBr_2_ + CsI solution (1.5 M in DMSO)]/PBT1-C/-C/MoO_3_/Ag**	1030	22.10	0.79	19.06	[162]
**ITO/SnO_2_/[CH_3_NH_3_PbI_3_]/PCDTBT/Ag**	970	19.90	0.73	14.08	[163]
**FTO/SnO_2_/[0.001 M MAI,0.001 M PbI_2_ + EACl (0.0002 M (15% molar ratio))]/PC3/Au**	1110	23.50	0.80	20.80	[164]
**ITO/SnO_2_/[1.1 M PbI_2_, 1.0 M FAI, 0.22 M PbBr_2_, 0.2 M MABr + 1.5 M CsI]/PBDTT/MoO_3_/Ag**	1120	23.64	0.77	20.28	[165]
**FTO/b-TiO_2_/m-TiO_2_/[CH_3_NH_3_PbI_3_]/P(hPhDTP)/Ag**	960	20.82	0.79	15.71	[166]
**FTO/TiO_2_/[0.0006 M PbI_2_, 0.0001 M PbBr_2_, 0.0001 M MABr, 0.0005 M FAI]/P-TT-TPD/Au**	1040	21.68	0.73	16.82	[167]
**FTO/SnO_2_/[CH_3_NH_3_PbI_3_]/PBDT[2F]T/Ag**	1060	22.64	0.73	17.52	[168]
**ITO/SnO_2_/[1.1 M PbI_2_, 1.0 M FAI, 0.22 M PbBr_2_, 0.2 M MABr]/PBDB-Cz/MoO_3_/Ag**	1135	24.34	0.76	21.11	[169]
**ITO/SnO_2_/[0.26 M FAI,1.26 M PbI_2_, 1.08 M MAI, 0.14 M PbCl_2_]/P25NH/Ag**	1049	19.81	0.83	17.30	[170]
**ITO/SnO_2_/[(MA_0.8_FA_0.2_)Pb(I_0.93_Cl_0.07_)_3_]/P5NH/Ag**	1041	20.95	0.83	18.10	[171]

## Data Availability

Not applicable.

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
