# Peer review of "Polymers in High-Efficiency Solar Cells: The Latest Reports"

_polymers, 2022, doi:10.3390/polym14101946_

Round 1
Reviewer 1 Report
The current work entitled “Polymers in high efficiency solar cells: the latest reports” by Gnida et al. presented on the latest reports on polymeric materials used in photovoltaic solar cells. Three types of solar cells: dye-sensitized, bulk heterojunction and perovskite solar cells are presented. The role of polymers and polymers thin films are described and discussed. The manuscript is very well written, and review results presented clearly. The manuscript seems interesting for the readers of “Polymers” The proposed work can be accepted after addressing the following minor comments.
Comments
- Please provide the future perspectives of the work.
- Provide the challenges and limitations that are currently facing with the polymers in the high efficiency solar cells.
- Add some more information in the section “Polymers in dye-sensitized solar cells”
- Caption of Figure 2 seems confusing, please revise it.
- Section 3.1. heading is it “Polymers as donors materials” or “Polymers as donor materials”
- Revise the line 538. “Wang, Tan et al.” seems incorrect. Like this mistake at many places. Please thoroughly check in the whole manuscript
- Provide the abbreviations place, in many places they were used.
- In table 4, reference number 105 (last one) correct the name of the Fabrication methods.
- Check the grammatical errors throughout the manuscript.
- At the end mention which polymers are best for the dye-sensitized, bulk heterojunction and perovskite solar cells
Reviewer 2 Report
This review paper provides detailed information regarding the use of polymeric materials in third generation solar cells. The manuscript is well written. I have some minor suggestions:
- There are many review papers available on the related topic. The novelty and advance of this paper should be highlighted and illustrated in the introduction section.
- In the introduction section, some papers can be cited regarding the use of conjugated polymers in BHJ solar cells: "Core/Shell Conjugated Polymer/Quantum Dot Composite Nanofibers through Orthogonal Non-Covalent Interactions." Polymers 8.12 (2016): 408.; "Structurally Diverse Poly (thienylene vinylene)s (PTVs) with Systematically Tunable Properties through Acyclic Diene Metathesis (ADMET) and Postpolymerization Modification." Macromolecules 49.9 (2016): 3318-3327.
- Besides the use of tables to summarize device performance, I suggest the authors to include some characterization results, such as SEM image, J-V curve of the device from representative literature. For example, section 4.1, SEM images can be presented to show how polymers improved the perovskite morphology.
- Table 9, the chemical composition of perovskite used in each device should be specified.
- I suggest the authors to include a future outlook section within the manuscript.
